# Heritable DNA methylation marks associated with susceptibility to breast cancer

Jihoon E. Joo[1,2], James G. Dowty[3], Roger L. Milne [3,4], Ee Ming Wong[1,2], Pierre-Antoine Dugué[3,4], kConFab, Dallas English[3,4], John L. Hopper[3], David E. Goldgar[1,5], Graham G. Giles[3,4] & Melissa C. Southey[1,2]

Mendelian-like inheritance of germline DNA methylation in cancer susceptibility genes has been previously reported. We aimed to scan the genome for heritable methylation marks associated with breast cancer susceptibility by studying 25 Australian multiple-case breast cancer families. Here we report genome-wide DNA methylation measured in 210 peripheral blood DNA samples provided by family members using the Infinium HumanMethylation450. We develop and apply a new statistical method to identify heritable methylation marks based on complex segregation analysis. We estimate carrier probabilities for the 1000 most heritable methylation marks based on family structure, and we use Cox proportional hazards survival analysis to identify 24 methylation marks with corresponding carrier probabilities significantly associated with breast cancer. We replicate an association with breast cancer risk for four of the 24 marks using an independent nested case–control study. Here, we report a novel approach for identifying heritable DNA methylation marks associated with breast cancer risk.

[1] Department of Pathology, The University of Melbourne, Melbourne, VIC 3010, Australia. [2] Precision Medicine, School of Clinical Sciences at Monash Health, Monash University, Clayton, VIC 3168, Australia. [3] Centre for Epidemiology and Biostatistics, The University of Melbourne, Melbourne, VIC 3010, Australia. [4] Cancer Epidemiology and Intelligence Division, Cancer Council Victoria, Melbourne, VIC 3004, Australia. [5] Huntsman Cancer Institute, Salt Lake, UT 84112, USA. Jihoon E. Joo and James G. Dowty contributed equally to this work. A full list of consortium members appears at the end of the paper. Correspondence and requests for materials should be addressed to kConFab, M.C.S. (email: msouthey@unimelb.edu.au)
A full list of consortium members appears at the end of the paper.

D NA methylation is a breast cancer risk factor. Several genome-wide studies of DNA methylation have found evidence that global methylation levels measured in blood-derived DNA is associated with breast cancer risk for women in the general population, and for women from families at high genetic risk[1–3]. While increased global methylation is associated with a reduced risk, increased methylation levels within functional promoters have been associated with an increased risk of breast cancer[2,3].

Candidate gene approaches have been used to assess whether methylation at CpG islands of breast cancer susceptibility genes is associated with breast cancer risk. Women carrying germline mutations in *BRCA1* have a substantially elevated risk of breast cancer and their tumours typically have distinctive histological features[4–6]. We found that peripheral blood DNA methylation at the *BRCA1* promoter was associated with an estimated 3.5-fold (95% CI, 1.4–10.5) increased risk of breast cancer diagnosed before the age of 40 years[7]. Hansmann et al.[8] reported that 1.4% of 600 women from the German Consortium for Hereditary Breast and Ovarian Cancer had constitutive *BRCA1* hyper-methylation confined to one of the two alleles[8].

Women carrying specific rare germline mutations in *ATM* are also at substantially elevated risk of breast cancer[9–11]. Flanagan et al.[12] performed methylation microarray analyses of peripheral blood DNA across several genes including *BRCA1*, *BRCA2*, *CHEK2*, *ATM*, *TP53*, *CDH1*, and *MLH1*, and demonstrated that gene body hypermethylation of *ATM* was associated with an estimated threefold increased risk of breast cancer[12]. Brennan et al.[13] combined two nested case–control studies of women at high risk of breast cancer and found evidence that methylation at an intragenic locus in *ATM* (ATMmvp2a) was associated with increased risk of breast cancer[13].

Potapova et al.[14] described promoter region methylation of *PALB2* was evident in ~7% of breast and ovarian cancers, including those with germline mutations in *BRCA2*, using methylation-specific PCR and bisulfite sequencing[14]. In contrast, Mikeska et al.[15] found little evidence of *PALB2* methylation in high-grade serous ovarian cancers using a methylation-sensitive high-resolution melting assay[15].

The terminology being used to describe these observations is variable and vulnerable to misuse and misinterpretation. The term 'epimutation' is strictly defined as a heritable change in gene activity that is not associated with a DNA mutation but rather with gain or loss of DNA methylation or other heritable modification of chromatin[16]. Changes in gene expression through altered DNA methylation or histone modifications induced from *cis*- or *trans*-acting genetic factors known as methylation Quantitative Trait Loci, (mQTL) are therefore not epimutations in this strict sense.

Epimutations and mQTLs can mimic germline mutations in their effect on cancer predisposition and it is likely that their contribution has been largely underestimated due to limited research beyond the candidate gene approaches described above[8]. These phenomena could therefore account for some of the familial risk of breast cancer that is not yet identified.

Intergenerational transmission of epimutations (as described by the authors in the initial reports) has been observed in *MLH1* and *MSH2* in the context of Lynch Syndrome (LS), a hereditary condition in which genetic mutations in key mismatch repair genes predispose individuals to colorectal, endometrial, and other cancers[17]. While two thirds of LS cases carry germline genetic mutations at the DNA mismatch genes[18], a small proportion of LS has been associated with epimutations[19,20]. It has since been demonstrated that some methylation marks at *MLH1* and *MSH2* that are transmitted transgenerationally are in fact linked to nearby *cis*-acting genetic variants and consequently follow

Mendelian inheritance patterns[21,22], and are thus not strictly epimutations. Other *MLH1* epimutations occur sporadically and have not been linked to underlying genetic variations[23]; while these epimutations are often observed in a familial context, they do not follow complete Mendelian inheritance patterns[23].

We hypothesised that breast cancers in multiple-case breast cancer families with no known genetic susceptibility mutations are in part due to the contribution of heritable DNA methylation marks (including true epimutations and mQTLs). To test this, we assessed genome-wide DNA methylation for 25 multiple-case breast cancer families using the Infinium HumanMethylation450 K BeadArray. One or more women with breast cancer in these families had been previously screened for, and found not to carry germline mutations in known breast cancer susceptibility genes. In this study, we report a new analytic approach to identify CpG sites with Mendelian-like inheritance patterns and a set of 24 heritable methylation sites associated with breast cancer risk.

## Results

**DNA methylation within families.** After removing 3949 poorly performing CpG probes (detection $p$-value < 0.05), $\beta$-values and $M$-values were obtained from a total of 481,563 analysable CpG probes across DNA samples from 210 individuals in 25 families (20 families participating in kConFab and 5 families participating in the ABCFR). β-values denote % methylation levels obtained from the HM450K platform, where 0 indicates 0% methylation and 1 indicates 100% methylation. Due to the heteroscedastic nature of $\beta$-values, the $\log_2$ ratio of methylation intensity, known as $M$-values, are also calculated and used for all statistical analyses[24].

DNA samples were collected from 87 breast cancer cases (one third of the cases had blood collected prior to diagnosis) and 123 unaffected controls. In order to examine the overall genome-wide methylation similarities between samples and families, a hierarchical clustering analysis was performed according to $M$-values across 481,563 probes. No distinct clustering by case–control status was observed but some families shared similar overall methylation patterns (Supplementary Fig. 1).

**Heritable methylation sites.** The proportion of probes within 10 bp of known single-nucleotide polymorphisms (SNPs) increased significantly with $\Delta l$ ($p < 0.0001$, and see Fig. 1). We then removed all probes within 10 bp of known SNPs and those located on sex chromosomes (see Methods). We screened the remaining 365,169 sites for those most consistent with having a Mendelian pattern of inheritance using the statistic $\Delta l$ (Supplementary Fig. 2A). The 1000 most Mendelian methylation marks (those with the highest values of $\Delta l$) are listed in Supplementary Data 1. These marks all have values of $\Delta l$ above 77, which suggests that they are highly heritable. We estimated carrier probabilities for the 1000 most heritable methylation marks using family structure alone.

**Heritable methylation sites associated with breast cancer.** Of the 1000 most Mendelian methylation marks, 24 of them had carrier probabilities that were associated with breast cancer at the Bonferroni-adjusted $p$-value threshold of $5 \times 10^{-5}$ (all $p$-values between $2 \times 10^{-5}$ and $7.4 \times 10^{-10}$, see Table 1 and Supplementary Fig. 2B). Notably, five of the heritable methylation marks were clustered together at *VTRNA2-1*. For all 24 marks, the methylation ($\beta$) differences were substantial ($\Delta\beta > 0.30$) between individuals, with most of these marks showing methylation patterns distinctly falling into hypermethylated ($\beta > 0.80$), hypomethylated ($\beta < 0.20$), or hemimethylated ($\beta \sim 0.50$) groups, indicating

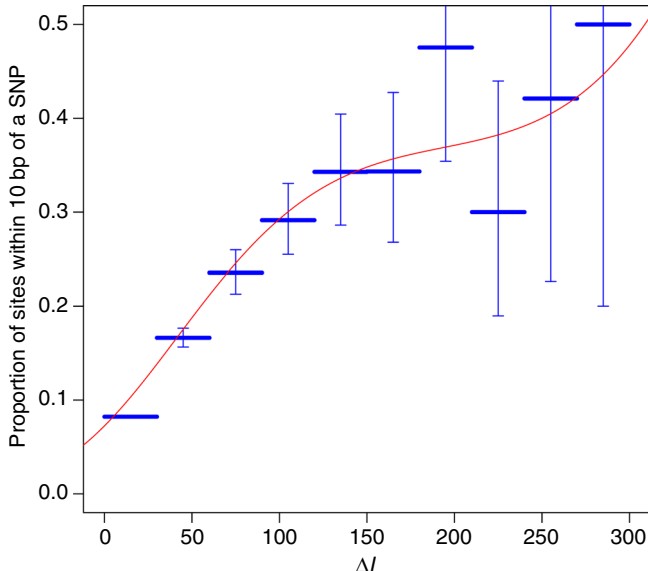

**Fig. 1** Predicting genetic variation with $\Delta l$. The proportion of methylation sites with nearby SNPs as a function of $\Delta l$, both by categories of $\Delta l$ (horizontal lines with error bars representing 95% confidence intervals) and as a polynomial function fitted by logistic regression (curvilinear line)

potential allele-specific methylation pattern at these sites (Supplementary Fig. 3 and Supplementary Table 1). While unbiased hazard ratios could not be calculated (see Statistical Methods), the risk of breast cancer increased with carrier probabilities for all 24 sites (Supplementary Table 2, where the low carrier probabilities for some CpGs reflect the very low prior carrier probability), and the estimated effect of the hypothetical genetic variant on the $M$-values of each site can be seen from Supplementary Fig. 3 and Supplementary Table 2. For example, for cg06536614, 'carriers' are hemimethylated and 'non-carriers' are hypomethylated. In contrast, for cg18584561, 'carriers' are hypomethylated and 'non-carriers' have generally higher methylation levels but these are spread over a range of methylation.

The five probes associated with the *VTRNA2-1* locus (previously known as *miR886*) encompass a ~50 bp region 150 bp upstream from the transcription start site and overlapping a CpG island. Although these probes target 5 independent proximally located CpG sites, the 50mer probes largely overlapped with each other (Fig. 2). In addition to these 5 CpG sites, DNA methylation at other proximal CpG probes showed similar patterns, although not meeting statistical significance. Within each individual, the methylation patterns at all CpGs across this *VTRNA2-1* promoter region were consistent, suggesting allelic methylation at this locus (Fig. 2b). Two recent studies have suggested that this region might be maternally imprinted[25,26]. We have tested this in eight trios (father, mother, and child) and included additional siblings when possible by performing clonal bisulfite sequencing. We observed strong hypermethylation of the maternally inherited allele, confirming the maternal imprinting of this locus. We found complete loss of methylation in one child whose three other siblings retained the methylation in the maternal allele (Supplementary Fig. 4).

One of the heritable methylation marks associated with breast cancer risk was located close to the 5′ end of the gene Growth Regulation by Estrogen Breast Cancer 1 (*GREB1*). The methylation patterns of all samples at this methylation site grouped clearly into hypomethylated, hemimethylated, or hypermethylated. Only 13 of our 210 samples were hypermethylated at this methylation site. We found three of the other methylation marks

overlapping promoter regions of *DUSP22*, *TMC3*, and *PPP2R5C*. Nine other heritable methylation marks were located in gene body regions of *MMP27*, *ANO10*, *CLGN*, *ZZEF1*, *PNKD*, *XYLT1*, *c7orf50*, *RASA3*, and *IL10RB*, while six heritable methylation marks were not known to be associated with any gene (Table 1). The *ZZEF1*, *PNKD*, *c7orf50*, *RASA3*, and *IL10RB* probes overlapped CpG island shores or shelves. The *MMP27*, *ANO10*, *XYLT1*, and *GREB1* probes encompassed enhancer regions.

**Breast cancer risk association in the general population**. Altogether, 433 invasive breast cancer cases and their matched controls were included in the analysis[2]. The median follow-up time was 9.5 years, interquartile range (IQR): 5.0 to 13.1 years. Supplementary Fig. 5 shows $\beta$-methylation value distribution for MCCS cases and controls for the 24 methylation sites showing heritable methylation patterns and associated with breast cancer in the family-based analyses. Of the 24 sites, four showed linear association with risk of breast cancer in the MCCS at the nominal significance threshold $p < 0.05$ (Table 2). The significant CpG probes were cg18584561 (*GREB1*; OR per standard deviation (s. d.): 1.18, 95% CI: 1.03–1.36), cg01741999 (*PNKD*; OR per 1 s.d.: 1.26, 95% CI: 1.03–1.54), cg03916490 (*C7orf50*; OR per 1 s.d.: 0.83, 95% CI: 0.72–0.96) and cg27639199 (*TMC3*; OR per 1 s.d.: 1.19, 95% CI: 1.03–1.36).

When comparing values belonging to the smaller vs. larger 'peak' of the methylation variable distribution, the results were consistent and more significant (Table 3). At cg18584561 (*GREB1*), which was trimodal, both the hypomethylated and hypermethylated peaks were associated with decreased breast cancer risk (OR = 0.60 (95% CI: 0.45–0.80), and OR = 0.56, (95% CI: 0.34–0.95), respectively). The methylation pattern at cg27639199 (*TMC3*) was also trimodal where the hypermethylated peak was strongly associated with breast cancer risk (OR = 2.16 (95% CI: 1.26–3.72)). At cg03916490 (*C7orf50*), reduced methylation was associated with the breast cancer risk (OR = 1.61 (95% CI: 1.16–2.24)). An annotated CpG probe (cg18514595) was associated with breast cancer risk when categorised into three methylation peaks.

These associations were robust to further adjustment for Houseman's white blood cell composition, and to further adjustment for additional breast cancer risk factors (parity, hormonal replacement therapy use, age at menarche and menopausal status). Similar results were also found when restricting the analyses to DNA that was extracted from dried blood spots (Supplementary Table 3) and when repeating the analyses with carrier probabilities in place of $M$-values (Supplementary Table 4).

**Associations between genetic variants and DNA methylation**. Genotyped and imputed variants from the iCOGS ($\pm 1$ kb of cg18584561, *GREB1*) representing 251 MCCS participates was included in the analysis. This region had eight common variants (in linkage disequilibrium) nominally associated with breast cancer risk. We found a very strong linear association between methylation at cg18584561 and the genotypes at this region ($p = 1 \times 10^{-65}$–$1 \times 10^{-71}$). The association between these genetic variants and the corresponding methylation $\beta$-value is presented graphically in Supplementary Fig. 6.

**Association with breast cancer estrogen receptor status**. We tested whether methylation levels at any of these 24 CpG sites were influenced by ER status in our nested case–control study and found evidence for three methylation marks cg06536614 (ER-; OR = 1.02 (95% CI: 0.86–1.21) vs. ER + : OR = 0.71 (95% CI: 0.53–0.96), $p$-value (heterogeneity) = 0.03), cg01074083 (ER-; OR

**Table 1 The methylation marks associated with breast cancer**

| CpG site | $\Delta I$ | P-value for association with breast cancer | Chromosome | Position (hg19) | UCSC reference gene |
|---|---|---|---|---|---|
| cg06536614 | 143.6285 | $7.23 \times 10^{-09}$ | 5 | 135416381 | VTRNA2-1 (MIR886) |
| cg10306192 | 109.4419 | $3.46 \times 10^{-05}$ | 11 | 102576374 | MMP27 |
| cg18110333 | 108.7894 | $4.13 \times 10^{-10}$ | 6 | 292329 | DUSP22 |
| cg00124993 | 107.9848 | $1.71 \times 10^{-08}$ | 5 | 135416412 | VTRNA2-1 (MIR886) |
| cg26328633 | 107.4759 | $1.64 \times 10^{-08}$ | 5 | 135416394 | VTRNA2-1 (MIR886) |
| cg25340688 | 105.9031 | $2.73 \times 10^{-08}$ | 5 | 135416398 | VTRNA2-1 (MIR886) |
| cg18514595 | 95.25137 | $1.67 \times 10^{-07}$ | 22 | 49579968 | unannotated |
| cg26896946 | 92.07959 | $1.50 \times 10^{-09}$ | 5 | 135416405 | VTRNA2-1 (MIR886) |
| cg11035303 | 90.90393 | $1.74 \times 10^{-10}$ | 3 | 43465503 | ANO10 |
| cg23012654 | 89.75858 | $3.85 \times 10^{-05}$ | 14 | 97493395 | unannotated |
| cg26773954 | 88.76923 | $1.12 \times 10^{-06}$ | 13 | 111969980 | unannotated |
| cg22901919 | 87.59356 | $1.85 \times 10^{-06}$ | 4 | 141317067 | CLGN |
| cg04417708 | 85.02877 | $1.28 \times 10^{-08}$ | 17 | 4043867 | ZZEF1 |
| cg18584561 | 85.00000 | $9.30 \times 10^{-06}$ | 2 | 11682017 | GREB1 |
| cg11608150 | 82.61516 | $5.21 \times 10^{-07}$ | 5 | 135415948 | unannotated |
| cg01741999 | 81.77624 | $3.28 \times 10^{-09}$ | 2 | 219137824 | PNKD |
| cg01074083 | 80.41676 | $1.58 \times 10^{-05}$ | 16 | 17516862 | XYLT1 |
| cg02096220 | 80.35092 | $3.64 \times 10^{-07}$ | 4 | 129212177 | unannotated |
| cg03916490 | 79.70945 | $2.07 \times 10^{-08}$ | 7 | 1080558 | C7orf50 |
| cg27639199 | 79.52796 | $5.37 \times 10^{-06}$ | 15 | 81666528 | TMC3 |
| cg25188166 | 79.40458 | $4.90 \times 10^{-08}$ | 3 | 119420208 | unannotated |
| cg05865327 | 78.94414 | $1.65 \times 10^{-06}$ | 14 | 102274741 | PPP2R5C |
| cg23947138 | 77.34483 | $7.47 \times 10^{-10}$ | 13 | 114782778 | RASA3 |
| cg05187003 | 77.22616 | $1.50 \times 10^{-08}$ | 21 | 34641507 | IL10RB |

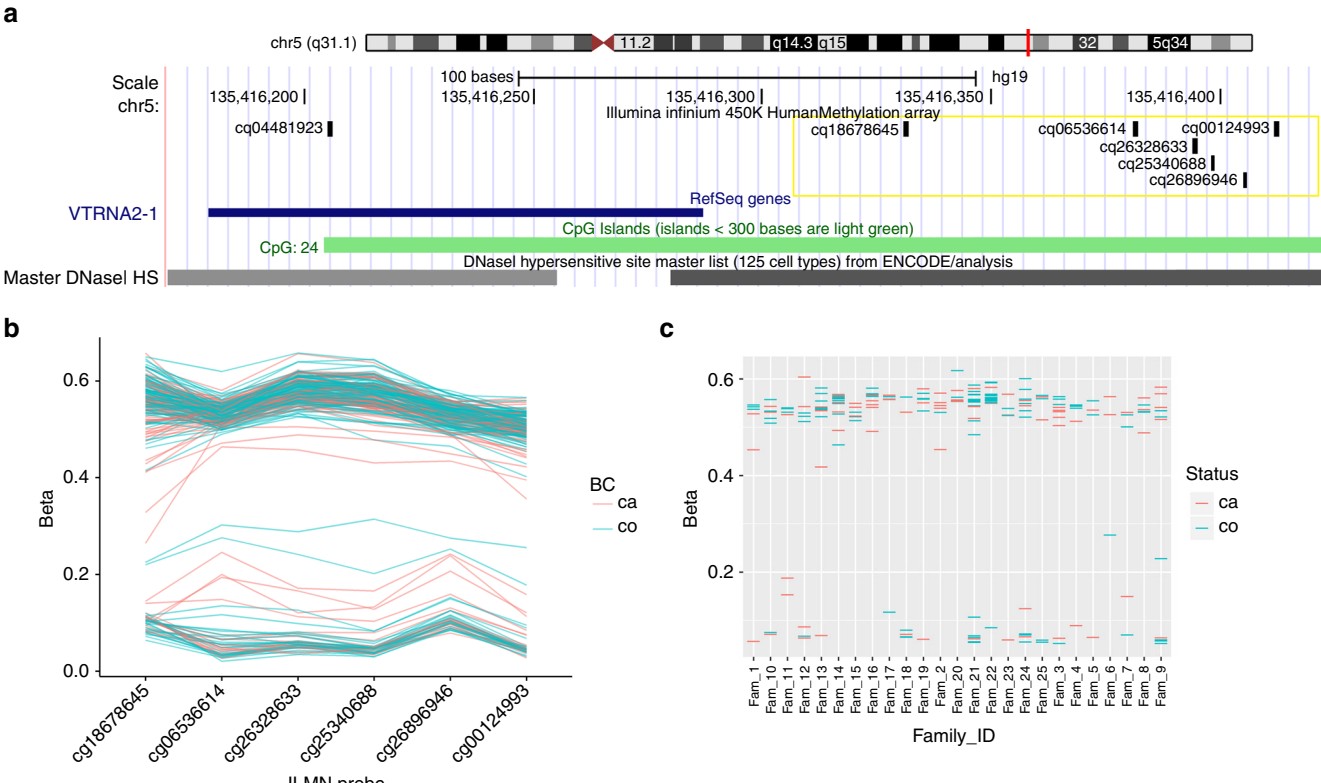

**Fig. 2** DNA methylation at the VTRNA2-1 promoter. **a** Genomic locations of 6 HM450K probes associated with *VTRNA2-1* promoter region. **b** DNA methylation levels (β-values) of these 6 probes labelled by breast cancer status. β-values for each individual are shown on y axis for 6 *VTRNA2-1* probes. **c** Average DNA methylation levels across all six probes shown separately for individual families and labelled by breast cancer status. β-values are shown on y axis for members from each family (y axis)

**Table 2 Associations between heritable DNA methylation marks (associated with breast cancer in multiple-case families) and risk of breast cancer in the general population (Melbourne Collaborative Cohort Study)**

| Site | Chr. | Position | Gene name | OR[a] | 95% CI | p |
|---|---|---|---|---|---|---|
| cg06536614 | 5 | 135416381 | *VTRNA2-1 (MIR886)* | 0.95 | 0.83–1.10 | 0.497 |
| cg10306192 | 11 | 102576374 | *MMP27* | 1.09 | 0.94–1.27 | 0.235 |
| cg18110333 | 6 | 292329 | *DUSP22* | 0.96 | 0.83–1.11 | 0.588 |
| cg00124993 | 5 | 135416412 | *VTRNA2-1 (MIR886)* | 0.97 | 0.84–1.12 | 0.667 |
| cg26328633 | 5 | 135416394 | *VTRNA2-1 (MIR886)* | 0.98 | 0.85–1.13 | 0.761 |
| cg25340688 | 5 | 135416398 | *VTRNA2-1 (MIR886)* | 0.95 | 0.82–1.09 | 0.441 |
| cg18514595 | 22 | 49579968 | *unannotated* | 1.14 | 0.99–1.31 | 0.077 |
| cg26896946 | 5 | 135416405 | *VTRNA2-1 (MIR886)* | 0.94 | 0.82–1.09 | 0.426 |
| cg11035303 | 3 | 43465503 | *ANO10* | 1.01 | 0.88–1.16 | 0.894 |
| cg23012654 | 14 | 97493395 | *unannotated* | 0.95 | 0.83–1.10 | 0.503 |
| cg26773954 | 13 | 111969980 | *unannotated* | 1.02 | 0.88–1.17 | 0.813 |
| cg22901919 | 4 | 141317067 | *CLGN* | 0.91 | 0.78–1.06 | 0.224 |
| cg04417708 | 17 | 4043867 | *ZZEF1* | 1.00 | 0.87–1.15 | 0.989 |
| **cg18584561** | **2** | **11682017** | ***GREB1*** | **1.18[b]** | **1.03–1.36** | **0.015** |
| cg11608150 | 5 | 135415948 | *unannotated* | 0.93 | 0.80–1.07 | 0.311 |
| **cg01741999** | **2** | **219137824** | ***PNKD*** | **1.26** | **1.03–1.54** | **0.027** |
| cg01074083 | 16 | 17516862 | *XYLT1* | 0.98 | 0.84–1.13 | 0.749 |
| cg02096220 | 4 | 129212177 | *unannotated* | 1.02 | 0.89–1.18 | 0.743 |
| **cg03916490** | **7** | **1080558** | ***C7orf50*** | **0.83** | **0.72–0.96** | **0.012** |
| **cg27639199** | **15** | **81666528** | ***TMC3*** | **1.19** | **1.03–1.36** | **0.018** |
| cg25188166 | 3 | 119420208 | *unannotated* | 0.96 | 0.83–1.10 | 0.551 |
| cg05865327 | 14 | 102274741 | *PPP2R5C* | 1.04 | 0.90–1.20 | 0.589 |
| cg23947138 | 13 | 114782778 | *RASA3* | 0.89 | 0.78–1.02 | 0.091 |
| cg05187003 | 21 | 34641507 | *IL10RB* | 1.00 | 0.86–1.15 | 0.950 |

[a] Odds ratio from conditional logistic regression of the risk of breast cancer on M-values (per 1 s.d.), adjusting for body mass index, tobacco smoking, alcohol drinking, time between blood collection and cancer diagnosis, and sample type (dried blood spots, peripheral blood mononuclear cells, and buffy coats). Cases and controls were individually matched on year of birth, year of blood draw, country of birth, and sample type for the vast majority of them (97%)
[b] Results are presented here using the methylation values as continuous, although the association was not linear. A better model fit was obtained by categorising into hypo/hemi/hypermethylated groups (i.e., peaks). Bold text indicates statistically significant associations

= 1.08 (0.90–1.29) vs. ER + : OR = 0.72 (0.53–0.98), *p*-value = 0.02) and cg23947138 (ER-: OR = 0.80 (0.68–0.94) vs. ER + : OR = 1.24 (0.93–1.65), *p*-value = 0.01). This result is shown in Supplementary Table 3.

## Discussion

Genome-wide studies of heritable DNA methylation studies in the context of familial breast cancer have not been conducted previously, although ~50% of familial breast cancer cases cannot be explained by what we currently know about genetic risk[27]. In this study, we tested whether heritable DNA methylation marks are associated with breast cancer risk in multiple-case breast cancer families that do not carry pathogenic mutations in known breast cancer susceptibility genes.

The hierarchical clustering analysis of all detected probes demonstrated that genome-wide methylation patterns were similar within some families, indicating that shared genetics might have an influence on DNA methylation, as shown in previous studies[28]. However, overall genome-wide methylation did not appear to segregate with affected status in any families (Supplementary Fig. 1).

We developed a new statistical methodology, based on an expectation–maximisation algorithm and genetic segregation analysis, to identify heritable DNA methylation marks using the HM450 K platform (see Methods). We validated this analytic approach by showing that it identifies probes that are known to overlap SNPs (the methylation measurements at these probes are likely to be influenced by the underlying SNPs). We then removed all SNP-overlapping probes from the analysis, screened the remaining probes for those with the most Mendelian-like inheritance patterns and tested some of the most heritable methylation marks for association with breast cancer. Note that our screening for probes with Mendelian-like inheritance patterns removed many probes that cannot be associated with familial breast cancer, so this screening step greatly increased our statistical power for detecting probes associated with familial breast cancer.

We found 24 probes associated with breast cancer risk after adjusting for multiple testing (Table 1). Five of these 24 CpG probes were adjacently located at the promoter region of a vault RNA, *VTRNA2-1* (previously known as *nc886* or *miR886*; Fig. 2). This vault RNA has been shown to be involved in the inhibition of protein kinase R (PKR) activity[29] and acts as a tumour suppressor in several cancer types[29–32]. It is located at chromosome 5q13, which is often associated with cancer-associated LOH including basal-like breast cancers[33,34]. Hypomethylation at this promoter, suggestive of loss of imprinting, occurs systematically in specific individuals in diverse populations, at least partially due to periconceptional environment and is stable for at least 10 years[35]. Silver et al. (2015) also noted that *VTRNA2-1* exhibits all the hallmarks of 'metabolic imprinting' and is likely to be a determinant of cancer risk[35]. Here we have shown that methylation at the *VTRNA2-1* promoter is also associated with heritable breast cancer risk that is measurable in DNA extracted from blood.

All 210 DNAs included in this study had hemi- or hypomethylation across all CpG probes at the *VTRNA2-1* locus (Fig. 2) indicating potential allele-specific DNA methylation (ASM). ASM at this locus has been reported previously by studies utilising clonal bisulfite sequencing of multiple tissue types[25,26]. However, these studies did not explore nearby genetic variation that could be superimposed on imprinting to influence the allelic methylation pattern. Hemimethylation patterns generally associated with genomic imprinting were only observed in 170 of the 210 DNAs (~80%) included in our study (Fig. 2). Genomic

**Table 3 Associations between heritable DNA methylation marks (associated with breast cancer in 608 multiple-case families) and risk of breast cancer in the general population (Melbourne Collaborative Cohort Study), *M*-values categorised into 2 or 3 groups according to observed bimodal or trimodal 610 distribution (i.e., peaks)**

| Site | Chr. | Position | Gene name | Smaller peak definition | *N* cases/controls in peak | OR[a] | 95% CI | *p* |
|---|---|---|---|---|---|---|---|---|
| cg06536614 | 5 | 135416381 | *VTRNA2-1* (MIR886) | $M < -1.8$ | 98/90 | 1.12 | 0.80–1.57 | 0.503 |
| cg10306192 | 11 | 102576374 | *MMP27* | $M > -2.5$ | 186/172 | 1.13 | 0.84–1.51 | 0.420 |
| cg18110333 | 6 | 292329 | *DUSP22* | $M < -2$ | 97/92 | 1.09 | 0.77–1.54 | 0.626 |
| cg00124993 | 5 | 135416412 | *VTRNA2-1* (MIR886) | $M < -2.8$ | 96/85 | 1.17 | 0.83–1.64 | 0.379 |
| cg26328633 | 5 | 135416394 | *VTRNA2-1* (MIR886) | $M < -2$ | 100/91 | 1.15 | 0.82–1.61 | 0.414 |
| cg25340688 | 5 | 135416398 | *VTRNA2-1* (MIR886) | $M < -2$ | 100/83 | 1.12 | 0.80–1.57 | 0.521 |
| **cg18514595** | **22** | **49579968** | ***Unannotated*** | $\mathbf{-2 < M < 2}$ | **183/156** | **1.35** | **1.00–1.82** | **0.048** |
| | | | | $\mathbf{M > 2}$ | **30/28** | **1.06** | **0.61–1.83** | **0.842** |
| cg26896946 | 5 | 135416405 | *VTRNA2-1* (MIR886) | $M < -1.5$ | 100/92 | 1.13 | 0.81–1.59 | 0.477 |
| cg11035303 | 3 | 43465503 | *ANO10* | $M > -2$ | 33/36 | 0.89 | 0.54–1.46 | 0.633 |
| cg23012654 | 14 | 97493395 | unannotated | $M < 2$ | 83/76 | 1.12 | 0.78–1.62 | 0.545 |
| cg26773954 | 13 | 111969980 | unannotated | $M < 2.2$ | 83/83 | 0.95 | 0.67–1.34 | 0.765 |
| cg22901919 | 4 | 141317067 | *CLGN* | $M < 1.5$ | 147/139 | 1.11 | 0.80–1.54 | 0.522 |
| cg04417708 | 17 | 4043867 | *ZZEF1* | $M < 2.5$ | 116/119 | 1.00 | 0.74–1.36 | 0.984 |
| **cg18584561** | **2** | **11682017** | ***GREB1*** | $\mathbf{M < -2}$ | **188/235** | **0.60** | **0.45–0.80** | **0.00045** |
| | | | | $\mathbf{M > 1}$ | **32/45** | **0.56** | **0.34–0.95** | **0.030** |
| cg11608150 | 5 | 135415948 | *Unannotated* | $M < -2.5$ | 118/102 | 1.22 | 0.88–1.67 | 0.229 |
| cg01741999 | 2 | 219137824 | *PNKD* | No peak | — | — | — | — |
| cg01074083 | 16 | 17516862 | *XYLT1* | $M < 2$ | 135/125 | 1.12 | 0.82–1.53 | 0.493 |
| cg02096220 | 4 | 129212177 | *Unannotated* | $M < 1.5$ | 151/153 | 1.03 | 0.77–1.38 | 0.825 |
| **cg03916490** | **7** | **1080558** | ***C7orf50*** | $\mathbf{M < 2.5}$ | **130/101** | **1.61** | **1.16–2.24** | **0.0047** |
| **cg27639199** | **15** | **81666528** | ***TMC3*** | $\mathbf{-1.5 < M < 1.5}$ | **189/181** | **1.23** | **0.92–1.64** | **0.157** |
| | | | | $\mathbf{M > 1.5}$ | **47/26** | **2.16** | **1.25–3.72** | **0.0059** |
| cg25188166 | 3 | 119420208 | *Unannotated* | $M < -0.5$ | 29/27 | 1.07 | 0.62–1.86 | 0.809 |
| | | | | $-0.5 < M < 1.5$ | 71/78 | 0.92 | 0.64–1.32 | 0.661 |
| cg05865327 | 14 | 102274741 | *PPP2R5C* | $M < 2.2$ | 106/106 | 0.95 | 0.68–1.32 | 0.760 |
| cg23947138 | 13 | 114782778 | *RASA3* | $M < 1.5$ | 115/95 | 1.33 | 0.97–1.82 | 0.075 |
| cg05187003 | 21 | 34641507 | *IL10RB* | No peak | — | — | — | — |

[a] Odds ratio from conditional logistic regression of the risk of breast cancer on *M*-values (per 1 s.d.), adjusting for body mass index, tobacco smoking, alcohol drinking, time between blood collection and cancer diagnosis, and sample type (dried blood spots, peripheral blood mononuclear cells, and buffy coats). Cases and controls were individually matched on year of birth, year of blood draw, country of birth, and sample type for the vast majority of them (97%)). Bold text indicates statistically significant associations (Table 2)

imprinting is usually highly effective and loss-of-imprinting is often associated with growth retardation syndromes or tumour development[36]. In reference to other typically imprinted region (e.g., *H19/IGF2*), the methylation at *VTRNA2-1* seemed exceptionally variable in the families included in our study. Romanelli et al. also report an atypical imprinting pattern at this locus and concluded this region was a polymorphically imprinted differentially methylated region[26]. By performing clonal bisulfite sequencing within families, we confirmed the polymorphic imprinting of this locus as reported by above studies (Supplementary Fig. 4).

We hypothesised that breast cancer arising in multiple-case breast cancer families with no known genetic mutation might be in part due to the contribution of heritable DNA methylation marks (including epimutations and mQTLs). As discussed above, methylation at the *VTRNA2-1* promoter is a strong epimutation candidate but many of the other identified heritable methylation marks are likely to be mQTLs. More work is required to characterise these marks further. It is not likely that common genetic variation currently recognised to be associated with breast cancer risk (already identified via genome-wide-association-studies) underlies these methylation marks. The currently published risk-associated SNP closest to any of the identified heritable methylation marks is located ~1.5b MB from cg18584561 (*GREB1*). We found a strong linear association between the DNA methylation

pattern at cg18584561 (*GREB1*) and 8 proximal common genetic variants (Supplementary Fig. 6). The genotypes of all 8 SNPs strongly correlated with the methylation pattern (e.g., DNAs hypermethylated at cg18584561 were homozygous across this region). This suggests a potential mQTL at this locus.

The other 19 CpG probes were all located in different genomic regions. We showed that a single CpG overlapping the transcription start region of the *GREB1* gene is associated with heritable breast cancer risk. This gene codes for the protein 'growth regulation by oestrogen in breast cancer 1' and has been shown to play a critical role in hormone dependent breast cancer[37,38]. There is currently no direct evidence of epigenetic regulation of this gene.

Four of the 24 methylation marks were associated with breast cancer risk in an independent nested case–control study of methylation and breast cancer risk (Table 2). This outcome provides information with which one could use to hypothesise further about the relative frequency of the 24 methylation marks. For some marks, such as the one at *GREB1*, approximately half of the families appear to be methylated which is consistent with replication being possible in a population-based sample, another fraction (~10%) of the population are hypermethylated at this CpG. Interestingly, both the hypomethylated and hypermethylated profiles were associated with a decreased breast cancer risk, with similar estimated risk reduction of 40–45%. At cg03916490

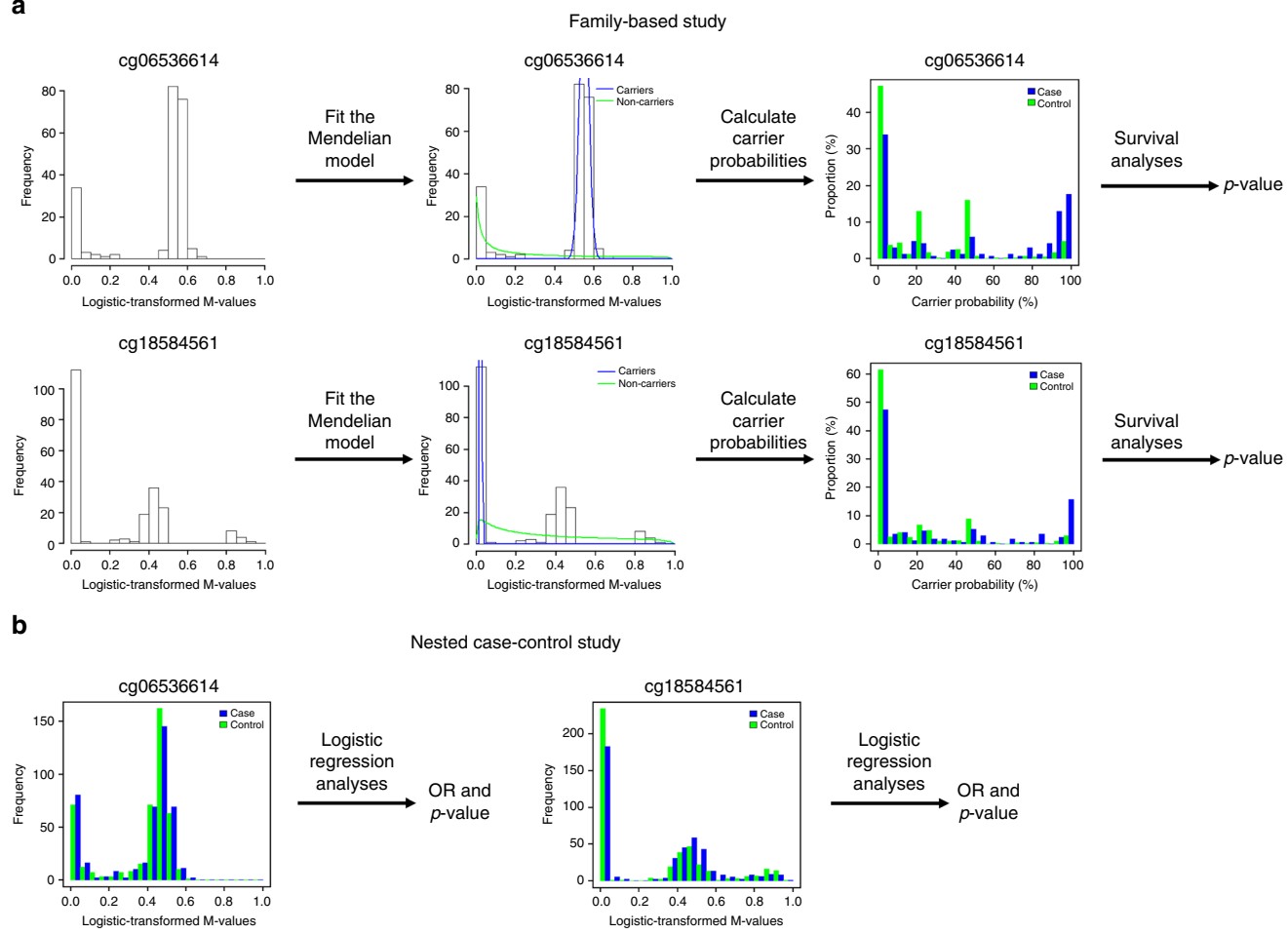

**Fig. 3** Analytical study approach. An overview of the analytical approach for each of the 1000 most-Mendelian probes in the multiple-case family-based analyses (**a**) and for the replication study of 24 probes in the population-based, case–control analyses (**b**). A measure of Mendelian heritability was calculated for all probes not on a sex chromosome or within 10 base pairs of a SNP (not depicted). For each of the 1000 most-Mendelian probes, a Mendelian model was fitted to the probe's *M*-values and this was used to calculate carrier probabilities (e.g., for a hypothetical genetic variant that causes aberrant DNA methylation at the probe), then these carrier probabilities were tested for association with breast cancer (note that unbiased *p*-values could be calculated but unbiased risks could not because we could not adjust for ascertainment). This gave 24 highly heritable methylation marks that were associated with breast cancer, and a nested case–control study was used to test the *M*-values of each of these probes for association with breast cancer and to estimate the corresponding odds ratios (ORs)

(*C7orf50*), about 30% of the nested case–control participants were not strongly hypermethylated, which was associated with a 60% increase in risk. It is possible that the marks that did not validate in the nest case–control sample were either not present or at a very low frequency. Fig 3 graphically illustrates our analytical approach using 2 CpGs with different 'carrier' probabilities as examples (cg06536614 and cg18584561).

Two thirds of the bloods collected from affected members of the multiple-case breast cancer families were collected after breast cancer diagnosis. Reverse causation is therefore a potential reason for non-replication of some of the methylation marks in the nested case–control study where blood samples were collected several years before breast cancer diagnosis.

Our study has two advantages over previous genome-wide studies. First, our approach utilises DNA methylation levels, which are important intermediate biomarkers that have not been incorporated into previous studies. Second, screening methylation marks for heritability is an effective way of greatly reducing the set of marks to test for association with breast cancer risk, but because all germline genetic variants are heritable by definition, this screening step could not be applied to previous studies.

Heritable methylation sites are interesting, regardless of whether or not they are associated with breast cancer susceptibility. We have devised a method for identifying heritable methylation sites and we have used this as a screening step to increase our power for detecting heritable methylation marks that are associated with breast cancer. This work could found a new area of exploration in the context of disease susceptibility. Specifically for breast cancer, this work provides new opportunities for increasing the precision of current risk prediction models, new strategies for cancer control (including screening) and new opportunities for the development of (or repurposing of) epigenetic therapeutics targeting these risk factors including chemo-prevention.

## Methods

**Study subjects**. Multiple-case breast cancer families. Subjects were members of 25 multi-generational families with multiple cases of breast cancer. The families were participants in the Kathleen Cunningham Foundation Consortium for Research into Familial Breast Cancer (kConFab) and the Australian Breast Cancer Family Registry (ABCFR)[39,40]. The present study was based on samples and phenotypic data from a total of 210 family members (87 affected and 123 unaffected) from 25 families and phenotypic data on their relatives.

One or more members of these families had undergone previous genetic testing and were not found to carry a mutation in a known breast cancer susceptibility gene. Genomic DNA was isolated from blood samples or (if no blood specimen was available) from Epstein-Barr virus transformed cell lines (Supplementary Data 2). All participants provided signed informed consent to participate in the relevant research resources. This study was approved by the Human Research Ethics Committee of The University of Melbourne (1441955) and meets the principles of the Declaration of Helsinki.

**Melbourne Collaborative Cohort Study (MCCS)**. Data from an independent nested case–control study of methylation as a risk factor for breast cancer within the Melbourne Collaborative Cohort Study (MCCS) were used to test the findings from the family analysis[41]. This included breast cancer cases with a first diagnosis of invasive adenocarcinoma of the breast (International Classification of Diseases for Oncology, C50) occurring between blood collection and 31 December 2007 and ascertained by record linkage to the population-based Victorian Cancer Registry (VCR), and to the Australian Cancer Database. Controls were selected through incidence density sampling and matched with cases on year of birth, year of baseline attendance, country of origin and, when possible, type of baseline blood specimen (dried blood spot, buffy coat, or lymphocyte). The HM450K array was used to measure genome-wide methylation in DNA prepared from peripheral blood sample collected prior to cancer diagnosis of the cases as described by Severi et al. (2014)[2]. All participants provided signed informed consent to participate in the relevant research resources.

**Bisulfite conversion and the HM450K array**. A total of 500 ng of genomic DNA per sample was bisulfite converted using Zymo Gold EZ-DNA kit (Irvine, CA). Prior to processing the bisulfite converted samples on the Infinium HM450 K BeadArray, the conversion was confirmed using bisulfite-specific PCR designed in-house[42]. The Infinium HM450 K (San Diego, CA) was performed using the TECAN automated liquid handler (Männedorf, Switzerland) according to the manufacturer's instruction.

**HM450K data processing**. All bioinformatic processing was performed with R version 3.2.0[43]. Raw intensity signals were imported and processed using the *minfi* package[44]. All samples had an average detection *p*-value < 0.001, indicating good quality data. Therefore, no sample was removed from the analysis. Wherever possible, individuals from the same families were run on the same chips. Individual CpG probes with detection *p*-value greater than 0.05 (3949 CpG probes) were deemed unreliable and excluded from further analyses. All samples were Illumina and SWAN normalised to reduce technical bias between Type 1 and Type 2 probes[45]. $\beta$-values and $M$-values were calculated in *minfi*[24,44]. $\beta$-values denote relative methylation percentage calculated from the ratio of the methylated probe intensity and the overall intensity, where 0 indicates 0% methylation and 1 indicates 100% methylation[24]. Due to the heteroscedastic nature of $\beta$-values and unsuitable for many statistical tests, $M$-values, which are the log2 of $\beta$-values, are also calculated[24]. Methylation measures from twelve technical duplicates were used for testing the reproducibility of methylation measures and removed from subsequent analysis. No further batch correction method was performed.

**Clonal bisulfite sequencing**. Clonal bisulfite sequencing was performed to test for the parent-of-origin allelic methylation patterns of the *VTRNA2-1* locus as previously described[25]. Germline DNA provided by 8 families, including 16 children were included in this analysis. All DNAs were first genotyped for rs2346019, (located at the downstream region of *VTRNA2-1*) using High-Resolution Melt curve analysis run on a RotorGene thermocycler (Qiagen, Hilden, Germany). Families where the allelic-specific methylation could be discriminated using this genotype information were selected for the bisulfite sequencing analysis (i.e., parents with disparate genotypes whose children were heterozygote at rs2346019). A set of previously published bisulfite-specific primers were used for amplifying the *VTRNA2-1* locus[25]. Cloning was performed using a TOPO-TA kit and at least 10 colonies per individual? were selected for Sanger Sequencing.

**Statistical methods**. Our method for identifying heritable methylation marks is based on a generalisation of the standard expectation–maximisation (EM) algorithm for Gaussian mixtures to allow for non-independent group memberships. These calculations were performed using custom code implemented in R version 3.1.1[43] because existing general segregation analysis software was too slow to make the calculations feasible for almost half a million probes.

For each methylation site (CpG probe), two statistical models were fitted to the site's $M$-values: a mixture model, in which the $M$-values were modelled as a mixture of two normal distributions (with means and variances to be estimated); and a Mendelian model, which is the same as the mixture model except that group membership was modelled as the carrier status (e.g., for a rare variant) at an autosomal genetic locus. Therefore, group memberships are independent under the mixture model but not under the Mendelian model. The maximised log-likelihoods, $l_{\text{mix}}$ and $l_{\text{Mendel}}$, for these models were calculated using the EM algorithm, with $l_{\text{mix}}$ obtained from the standard EM algorithm for Gaussian mixtures[46] and $l_{\text{Mendel}}$ calculated using the modification of this algorithm described

in The EM algorithm for the Mendelian model, below. For each model, setting the means and variances for the two groups to be equal corresponds to a Gaussian model in which the $M$-values follow a normal distribution, so this Gaussian model is nested inside both the mixture and Mendelian models. Using the likelihood ratio test to compare these models to the Gaussian model is uninformative because many probes appear to have a bimodal distribution, so instead we compared $l_{\text{mix}}$ to $l_{\text{Mendel}}$. A maxim from the field of statistical model selection is that the maximised log-likelihood quantifies how well a model fits the observed data[47]. Therefore, $\Delta l = l_{\text{Mendel}} - l_{\text{mix}}$ is a measure of how Mendelian the probe's $M$-values are, over and above how bimodal they are. Note also that since the mixture and Mendelian models have the same number of model parameters, $\Delta l$ is the difference between the AICs for these two models, so the AIC model-selection approach would select the Mendelian model in preference to the mixture model whenever $\Delta l > 0$ (and similarly for the BIC)[47].

To validate the ability of the $\Delta l$ statistic to identify methylation sites with Mendelian-like inheritance patterns, we calculated $\Delta l$ for all 481,563 methylation sites and used logistic regression and the likelihood ratio test to test whether or not the proportion of probes within 10 bp of a known SNP increases with $\Delta l$. This is a test on the efficacy of our statistic $\Delta l$, because the observed $M$-values of methylation probes with nearby SNPs are likely to have Mendelian-like inheritance patterns, just as an artefact of how the HM450 array measures methylation[48]. The HM450K probes are 50mer oligonucleotides in design with the interrogated target CpGs at the last base. A technical limitation of the platform is that a large proportion of probes overlap one or more known SNPs[48]. As the accuracy of methylation measurements relies on the efficient hybridising of probes to target complementary DNA fragment, SNPs within probes potentially interfere with this binding and interrupt the actual methylation measurements[48]. The observed methylation values are therefore biased by nearby SNPs and will tend to follow Mendelian patterns of inheritance. We could therefore assess if $\Delta l$ identified heritable sites by testing whether probes with higher values of $\Delta l$ were more likely to have nearby SNPs. In addition to the formal test above, we also binned probes by their values of $\Delta l$ and graphed the proportion of probes within 10 bp of a known SNP for each bin. Known SNPs were defined by Illumina's HM450 K Manifest v1.2 (see Web resources).

To identify heritable methylation marks associated with breast cancer, we first excluded all methylation probes on sex chromosomes or within 10 bp of known SNPs. Then we screened the remaining 365,169 probes for those most consistent with a Mendelian pattern of inheritance, using the statistic $\Delta l$. Note that this screening was based on the structure of the 25 families and did not use any data on breast cancer-affected status or age. For each of the 1000 most Mendelian sites (those with the highest values of $\Delta l$), we calculated carrier probabilities for the hypothetical genetic variant that determines group membership in the Mendelian model. These calculations used standard techniques from segregation analysis[49], in which the observed $M$-values played the role of the 'phenotypes' and the Gaussian densities (with the model parameters equal to their maximum likelihood estimates from the Mendelian model) played the role of the 'penetrance' function. The calculation of these carrier probabilities also only used pedigree structure and $M$-values, not age or breast cancer data.

Cox proportional hazards survival analysis was then used to test for associations between breast cancer and the carrier probabilities for the 1000 most Mendelian methylation marks. These analyses were conducted in R version 3.1.1[43] using the *coxph* function of the survival package[50]. To adjust for multiple testing, a Bonferroni-corrected *p*-value threshold of 0.05/1000 was used to determine statistical significance. Note that the effects of multiple testing were greatly reduced in our study because we screened the methylation sites for those with Mendelian inheritance patterns before testing for association with breast cancer.

The families in this study were ascertained because they each contained multiple breast cancer cases, and no adjustment for this ascertainment criterion was made. This means that our hazard ratio estimates are biased, so we do not report these here, but since the ascertainment criterion has no effect on the test statistic under the null hypothesis, our *p*-values for association with breast cancer are valid. These *p*-values were based on the likelihood ratio test, not the Wald test, so variances for the hazard ratios were not needed and hence were not estimated using either standard maximum likelihood or robust variance estimators.

**The EM algorithm for the Mendelian model**. This section gives a detailed, mathematical description of our generalization of the standard EM algorithm for Gaussian mixtures to allow for non-independent group memberships, as well as a precise description of the above statistic $\Delta \ell$ and its two related statistical models.

The statistic $\Delta l$ for measuring how Mendelian the inheritance pattern of a given site is: for each of the methylation sites, we fitted two statistical models to the sites' $M$-values $x_1, \ldots, x_n$, where $n$ is the number of people with epigenome-wide data and $x_i$ is the site's $M$-value for person $i$. The first model is a mixture of two Gaussians, so under this model there are binary random variables $y_1, \ldots, y_n$ so that: the $n$ bivariate random variables $(x_1, y_1), \ldots, (x_n, y_n)$ are independent; and for each $j = 0$ or 1, $P(y_i = j) = \alpha_j$ and $(x_i | y_i = j) \sim N(\mu_j \sigma_j^2)$, where $\theta = (\alpha_0, \alpha_1, \mu_0, \mu_1, \sigma_0, \sigma_1)$ is a vector of parameters to be estimated while satisfying the constraint $\alpha_0 + \alpha_1 = 1$. In this paper, we will also impose the additional constraint that $\alpha_1 = 0.01$, so $\alpha_0$ and $\alpha_1$ are fixed constants. The second model is the same as the first, except that the group membership variables $y_1, \ldots, y_n$ are modelled as the carrier status for a rare, autosomal genetic variant, with $y_i = 1$ if individual $i$ is a carrier and $y_i = 0$ if he or

she is a non-carrier. Note that $y_i$ and $y_j$ will generally be dependent random variables if individuals $i$ and $j$ belong to the same pedigree, though we still assume that $x_1,\dots,x_n$ are conditionally independent given $y_1,\dots,y_n$.

We will refer to these models as the mixture and Mendelian models, respectively. Setting $\mu_0 = \mu_1$ and $\sigma_0 = \sigma_1$ in either of these models gives a third model for the $M$-values, in which $x_1,\dots,x_n$ are independent and follow a univariate normal distribution, that we call the Gaussian model. The maximised log-likelihoods $\ell_{\text{mix}}$, $\ell_{\text{Mendel}}$, and $\ell_{\text{Gauss}}$ of these three models measure the goodness-of-fit of each model to the site's $M$-values[47]. Since the Gaussian model is nested inside the other two models, $\ell_{\text{mix}}$ and $\ell_{\text{Mendel}}$ can both be formally compared to $\ell_{\text{Gauss}}$ using a likelihood ratio test in order to determine if either of these models gives a more parsimonious fit to the data than the Gaussian model. However, the $M$-values of a very large number of the sites are bimodal, so these tests very often prefer both of the other models to the Gaussian model. To discover sites whose methylation patterns are Mendelian, we therefore compare $\ell_{\text{Mendel}}$ to $\ell_{\text{mix}}$, even though the mixture and Mendelian models are not nested. Since these models have the same number of parameters, $\Delta\ell = \ell_{\text{Mendel}} - \ell_{\text{mix}}$ is the difference in both the AIC and BIC of the two models, so if $\Delta\ell > 0$ then the AIC and BIC would both favour the Mendelian model over the mixture model as the more parsimonious description of the data[47]. Also, since $\ell_{\text{mix}}$ and $\ell_{\text{Mendel}}$ measure the goodness-of-fit of these models to the site's $M$-values[47], the better the Mendelian model fits the data compared to the mixture model, the larger $\Delta\ell$ should be. We therefore interpret $\Delta\ell$ as a statistic which measures how 'Mendelian' the site is, i.e. how consistent the observed $M$-values at the site are with a Mendelian pattern of inheritance within families.

Note that we have assumed that all familial aggregation of aberrant DNA methylation is due to a major gene, so $\ell_{\text{Mendel}}$ and hence $\Delta\ell$ will be upwardly biased if part of this familial aggregation is caused by multiple genes of small effect (i.e., a polygenic effect), or if our model is misspecified in other ways. However, note that we only use $\Delta\ell$ to rank the methylation sites, and this ranking is completely insensitive to a wide range of biases. Also, while there are good theoretical and empirical reasons for using $\Delta\ell$ to screen the methylation sites, this screening is not a formal statistical procedure, so even if $\Delta\ell$ were biased then this would have no effect on the validity of our tests for association with breast cancer (the only formal part of our analysis). Finally, we note that replacing the Mendelian model with a mixed model (a model that incorporates a polygene in addition to a major gene) would possibly identify sites with polygenic but not Mendelian patterns of inheritance, which we are not interested in here.

A detailed description of the EM algorithm for the Mendelian model: since our analysis included approximately 480,000 sites, efficient algorithms were needed to maximise the likelihoods. For the mixture model this was straight-forward, because the EM algorithm for a mixture of Gaussians results in analytical update formulae[46], which can be iterated to rapidly converge (in most cases) to the maximum likelihood estimates. For the Mendelian model, we used a modification of this algorithm that we now describe in detail.

In the EM algorithm for the Mendelian model, we took the $M$-values $x_1,\dots,x_n$ of a given site as the observed data and the binary carrier statuses $y_1,\dots,y_n$ as the hidden data. For now, the reader can simply think of $y_1,\dots,y_n$ as variables defining group memberships, as in the standard EM algorithm for Gaussian mixtures[46], though with the caveat that $y_1,\dots,y_n$ are not independent. With model parameters $\theta = (\alpha_0,\alpha_1,\mu_0,\mu_1,\sigma_0,\sigma_1)$ as above, if $\theta_t$ is the estimate of these parameters at iteration $t$ then the EM algorithm chooses the estimate $\theta_{t+1}$ at the next iteration to be the argument which maximises the function of $\theta$ given by

$$Q(\theta, \theta_t) = \mathbb{E}[\log P(x, y|\theta)|x, \theta_t]$$

where $x = (x_1,\dots,x_n)$, $y = (y_1,\dots,y_n)$, $\mathbb{E}[\cdot]$ is the expectation functional and $P(x,y|\theta)$ is the likelihood of the full data at parameter value $\theta$. More precisely, if $\mathcal{Y} = \{0,1\}^n$ is the set of all binary vectors of length $n$, then

$$
\begin{aligned}
Q(\theta, \theta_t) &= \sum_{y\in\mathcal{Y}} P(y|x,\theta_t)\log P(x,y|\theta) \\
&= \sum_{y\in\mathcal{Y}} P(y|x,\theta_t)\log[P(x|y,\theta)P(y|\theta)] \\
&= \sum_{y\in\mathcal{Y}} P(y|x,\theta_t)\log P(x|y,\theta) + \sum_{y\in\mathcal{Y}} P(y|x,\theta_t)\log P(y|\theta) \\
&= \sum_{y\in\mathcal{Y}}\sum_{i=1}^{n} P(y|x,\theta_t)\log P(x_i|y_i,\theta) + \sum_{y\in\mathcal{Y}} P(y|x,\theta_t)\log P(y|\theta)
\end{aligned}
\tag{1}
$$

since the $M$-values $x_1,\dots,x_n$ are assumed to be conditionally independent given the carrier statuses $y$, with the distribution of $x_i$ only a function of $y_i$. The first sum in (1) is a function only of the parameters $\mu_0$, $\mu_1$, $\sigma_0$, and $\sigma_1$, while the second sum only depends on $\alpha_0$ and $\alpha_1$. So, to find $\theta$ that maximises $Q(\theta,\theta_t)$, we can maximise these two functions separately. In the analysis presented in this paper, however, $\alpha_0$ and $\alpha_1$ were fixed to the values 0.99 and 0.01, respectively, so we focus on maximising the first term of (1) here.

Let $\delta_{ij}$ denote the Kronecker delta, and for each $j = 0$ or 1, let $\varphi(x_i|\mu_j,\sigma_j)$ be the probability density function for the normal distribution $N(\mu j,\sigma^2)$ evaluated at $x_i$, so

that $P(x_i|y_i,\theta) = \phi(x_i|\mu_{yi},\sigma_{yi})$. Then the first sum in (1) is

$$
\begin{aligned}
&\sum_{y\in\mathcal{Y}}\sum_{i=1}^{n} P(y|x,\theta_t)\log P(x_i|y_i,\theta) \\
&= \sum_{y\in\mathcal{Y}}\sum_{i=1}^{n} P(y|x,\theta_t)\log\phi(x_i|\mu_{y_i},\sigma_{y_i}) \\
&= \sum_{y\in\mathcal{Y}}\sum_{i=1}^{n} P(y|x,\theta_t)\sum_{l=0}^{1}\delta_{ly_i}\log\phi(x_i|\mu_l,\sigma_l) \\
&= \sum_{i=1}^{n}\sum_{l=0}^{1} q_{il}^{t}\log\phi(x_i|\mu_l,\sigma_l)
\end{aligned}
\tag{2}
$$

where

$$q_{il}^{t} = \sum_{y\in\mathcal{Y}} \delta_{ly_i} P(y|x,\theta_t) \tag{3}$$

so that $q_{il}^{t}$ is the carrier probability for person $i$ corresponding to $x$ and the parameter values $\theta_t$ when $l = 1$ (note that the $t$ in $q_{il}^{t}$ is a general superscript, not a power). Therefore, (2) is a weighted log-likelihood of normal distributions, so it can be maximised in exactly the same way as for the standard EM algorithm for Gaussian mixtures[46]. This gives the following parameter values at iteration $t + 1$, for each $l = 0,1$:

$$\mu_l^{t+1} = \sum_{i=1}^{n} w_{il}^{t}x_i \text{ and } \sigma_l^{t+1} = \sqrt{\sum_{i=1}^{n} w_{il}^{t}\left(x_i - \mu_l^{t+1}\right)^2} \tag{4}$$

where $w_{il}^{t} = q_{il}^{t}/\sum_{j=1}^{n} q_{jl}^{t}$ and, as before, the superscripts $t$ and $t + 1$ are not exponents.

To calculate these estimates, we used the definition (3) of $q_{il}^{t}$ and the following expression for $P(y|x,\theta_t)$. Let $F$ be the partition of $\{1,\dots,n\}$ into families, so that $F$ is a set of sets of indices, with each $f\in F$ of the form $f = \{i_1,\dots,i_k\}$, where $i_1,\dots,i_k$ are all of the people in a given family with epigenome-wide methylation data. For any such $f\in F$, let $x^f = (x_{i1,\dots},x_{ik})$ and $y^f = (y_{i1,\dots},y_{ik})$ be the observed and hidden data for the family, respectively. Then since the carrier statuses and $M$-values of people from different families are independent,

$$P(y|x,\theta_t) = \prod_{f\in F} P(y^f|x^f,\theta_t) = \prod_{f\in F} \frac{P(x^f|y^f,\theta_t)P(y^f|\theta_t)}{P(x^f|\theta_t)} \tag{5}$$

To calculate $P(y|x,\theta_t)$ from the right-hand side of (5), we note that, as before,

$$P(x^f|y^f,\theta_t) = \prod_{i\in f} P(x_i|y_i,\theta_t) = \prod_{i\in f} \phi(x_i|\mu_{y_i}^t,\sigma_{y_i}^t)$$

Also, $P(y^f|\theta_t)$ can be calculated using standard techniques from segregation analysis[49], as described in more detail in Statistical methods, above. Finally, the denominator $P(x^f|\theta_t)$ in the right-hand side of (5), which is just a normalising constant, can be obtained by summing the numerator overall values of $y^f$. Therefore, $P(y|x,\theta_t)$ can be calculated from (5), so substituting this into (3) gives $q_{il}^{t}$ which, by (4), gives the updated parameters for the EM algorithm.

Improving calculation speeds: our analyses of ~480,000 sites would not be feasible without a number of techniques to improve the speed of the EM algorithm for the Mendelian model, so we briefly describe two of these techniques now.

The Mendelian model is a segregation analysis model[49], and for such models the most time-consuming part of the calculation is summing overall possible genotype combinations for all family members in each family. However, this part of the calculation is essentially common to all methylation sites, so we obtain considerable improvements in speed by performing this calculation once and storing the results for later use.

More precisely, the update equations (4) for the EM algorithm depend on the carrier probabilities $P(y^f|\theta_t)$ via (3) and (5), where we recall that $y^f$ is a set of carrier statuses for all of the members of family $f$ with epigenome-wide data. Using standard techniques from segregation analysis[49], $P(y^f|\theta_t)$ can be expressed as a sum over all genotype combinations for the family which are consistent with the genotypes $y^f$. Evaluating these sums is usually very time-consuming, however $P(y^f|\theta_t)$ depends on $\alpha_0^t$ and $\alpha_1^t$ but not on $\mu_1^t$, $\sigma_0^t$ or $\sigma_1^t$, and $\alpha_0^t$ and $\alpha_1^t$ are held fixed for all $t$, so $P(y^f|\theta_t)$ does not depend on $t$ or the $M$-values $x^f$. Therefore, we calculated $P(y^f|\theta_t)$ once for every possible combination $y^f$ of genotypes, and stored these values of $P(y^f|\theta_t)$ for later use in the update equations (4) (via (3) and (5)) for each methylation site.

We also used a simplifying assumption. To reduce the number of genotype combinations $y^f$ for which we had to store values of $P(y^f|\theta_t)$, we assumed that no more than 1 of the founders in each family is a carrier and that no founder is a homozygote carrier (as usually holds if the variant is rare). This assumption is not essential, however, and it can be weakened (e.g., to allow 2 variants or less among the alleles of the founders) or entirely dispensed with (if the families are not too large and not too many family members have epigenome-wide data).

**Testing 24 methylation marks in the MCCS.** For each of the 24 CpG sites of interest, we first estimated odds ratios (OR) for breast cancer risk using conditional logistic regression models, for a one standard deviation increase in the methylation $M$-values in blood HM450K data set of 433 cases and their matched controls from the MCCS. The models were adjusted for body mass index, tobacco smoking, alcohol drinking, time between blood collection, and cancer diagnosis, and sample type (DNA extracted from dried blood spots, peripheral blood mononuclear cells, and buffy coats, although the vast majority (97%) of case–control pairs were successfully matched on sample type). For methylation marks exhibiting a bimodal or trimodal distribution, we categorised the methylation variables into groups corresponding to the observed 'peaks' of hypo, hemimethylated or hypermethylated, based on visual inspection of the $M$-value distribution (Supplementary Fig. 4). We used the same models as for the continuous variable analyses. The larger peak was chosen as the reference category. Sensitivity analyses were conducted: (1) further adjusting the models for blood cell composition as estimated by the algorithm by Houseman et al.[51]; (2) further adjusting the models for age at menarche, menopausal status, number of live births, and use of hormonal replacement therapy; (3) restricting the analyses to DNA prepared from dried blood spots.

**Associations between genetic variants and DNA methylation.** Data for all variants with 1 kb of the GREB1 probe that were genotyped or imputed using the iCOGS array were retrieved for MCCS participants included in the Breast Cancer Association Consortium[52]. A total of 251 participants (231 cases and 20 controls) had iCOGS and HM450K data available. Association between genotype and methylation was assessed using linear regression, with beta-value as the outcome variable and allele count as the explanatory variable. The allele count was estimated by rounding the allele dose to an integer value.

**Web resources.** Illumina Infinium HumanMethylation450K manifest was downloaded from http://support.illumina.com/array/array_kits/ infinium_humanmethylation450_beadchip_kit/downloads.html

**Data availability.** All DNA methylation data (HM450K array) has been deposited to GEO (Accession No. GSE104942) and all bisulfite sequencing data has been deposited into BankIt2071934 (MG686237-MG686418) and is freely available.

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

## Acknowledgements

The Australian site of Breast Cancer Family Registry was supported by grant UM1 CA164920 from the USA National Cancer Institute. The content of this manuscript does not necessarily reflect the views or policies of the National Cancer Institute or any of the collaborating centres in the Breast Cancer Family Registry (BCFR), nor does mention of trade names, commercial products, or organisations imply endorsement by the USA Government or the BCFR. We thank Heather Thorne, Eveline Niedermayr, all the kConFab research nurses and staff, the heads and staff of the Family Cancer Clinics, and the Clinical Follow-Up Study (which has received funding from the NHMRC, the National Breast Cancer Foundation, Cancer Australia, and the National Institute of Health (USA)) for their contributions to this resource, and the many families who contribute to kConFab. kConFab is supported by a grant from the National Breast Cancer Foundation, and previously by the National Health and Medical Research Council (NHMRC), the Queensland Cancer Fund, the Cancer Councils of New South Wales, Victoria, Tasmania and South Australia, and the Cancer Foundation of Western Australia. We would like to express our gratitude to the many thousands of Melbourne residents who continue to participate in the Melbourne Collaborative Cohort Study, the original investigators, programme managers and the diligent team who recruited the participants and who continue working on follow-up. The MCCS methylation work was supported by the National Health and Medical Research Council (Grant number 1011618); and the Victorian Breast Cancer Research Consortium. M.C.S. is a Senior Research Fellow and J.L.H. is a Senior Principal Research Fellow of the National Health and Medical Research Council of Australia. This work was supported by an Early Career Research Award to JEJ from The University of Melbourne.

## Author contributions

This study was first conceived and designed by J.E.J., E.M.W. and M.C.S. J.E.J. performed all laboratory experiments. J.E.J., J.G.D., R.L.M. and P.A.D. performed bioinformatics and statistical analyses. R.L.M. and G.G.G. facilitated the inclusion and interpretation of the data from the MCCS. Study materials were provided by kConFab, ABCFR, and MCCS. The manuscript was first structured by J.E.J., J.G.D., and M.C.S. D.E., J.L.H. and D.E.G. provided significant intellectual contributions. All authors reviewed the manuscript.

## Additional information

**Competing interests:** The authors declare no competing interests.

## kConFab

Adrienne Sexton[6], Alice Christian[7], Alison Trainer[8], Allan Spigelman[9], Andrew Fellows[8], Andrew Shelling[10], Anna De Fazio[11], Anneke Blackburn[12], Ashley Crook[13], Bettina Meiser[14], Briony Patterson[15], Christine Clarke[16], Christobel Saunders[17], Clare Hunt[18], Clare Scott[19], David Amor[20], Deborah Marsh[21], Edward Edkins[22], Elizabeth Salisbury[23], Eric Haan[24], Eveline Niedermayr[25], Finlay Macrae[6], Gelareh Farshid[26], Geoff Lindeman[27], Georgia Chenevix-Trench[28], Graham Mann[29], Grantley Gill[30], Heather Thorne[8], Ian Campbell[8], Ian Hickie[31], Ingrid Winship[32], Jack Goldblatt[33], James Flanagan[34], James Kollias[35], Jane Visvader[19], Jennifer Stone[36], Jessica Taylor[6], Jo Burke[37], Jodi Saunus[38], John Forbes[39], Jonathan Beesley[28], Judy Kirk[40], Juliet French[28], Kathy Tucker[41], Kathy Wu[42], Kelly Phillips[43], Lara Lipton[44], Leslie Andrews[40], Elizabeth Lobb[45], Logan Walker[46], Maira Kentwell[6], Amanda Spurdle[28], Margaret Cummings[47], Margaret Gleeson[48], Marion Harris[49], Mark Jenkins[50], Mary Anne Young[51], Martin Delatycki[52], Mathew Wallis[53], Matthew Burgess[52], Melanie Price[54], Melissa Brown[55], Michael Bogwitz[6], Michael Field[56], Michael Friedlander[57], Michael Gattas[58], Mona Saleh[59], Nick Hayward[28], Nick Pachter[32], Paul Cohen[60], Pascal Duijf[61], Paul James[6], Peter Simpson[62], Peter Fong[63], Phyllis Butow[64], Rachael Williams[41], Richard Kefford[65], Rodney Scott[66], Rosemary Balleine[64], Sarah-Jane Dawson[8], Sheau Lok[67], Shona O'Connell[48], Sian Greening[68], Sophie Nightingale[8], Stacey Edwards[28], Stephen Fox[69], Sue-Anne McLachlan[70], Sunil Lakhani[71], Susan Thomas[72] & Yoland Antill[73]

[6]Familial Cancer Centre, Royal Melbourne Hospital, Grattan Street, Parkville, VIC 3050, Australia. [7]Genetics Department, Central Region Genetics Service, Wellington Hospital, Wellington 6021, New Zealand. [8]The Peter MacCallum Cancer Centre, Victorian Comprehensive Cancer Centre, Grattan Street, Melbourne 3000, Australia. [9]Family Cancer Clinic, St Vincents Hospital, Darlinghurst NSW 2010, Australia. [10]Obstetrics and Gynaecology, University of Auckland, Auckland 1010, New Zealand. [11]Dept. Gynaecological Oncology, Westmead Institute for Cancer Research, Westmead Hospital, Westmead, NSW 2145, Australia. [12]Australian National University, P.O. Box 334 Canberra, ACT 2601, Australia. [13]Department of Clinical Genetics, Level 3E, Royal North Shore Hospital, St Leonards, NSW 2065, Australia. [14]Prince of Wales Hospital, The University of New South Wales, UNSW, Sydney, NSW 2052, Australia. [15]Clinical Genetics Service, Royal Hobart Hospital, GPO Box 1061 Hobart, TAS 7001, Australia. [16]Westmead Institute for Cancer Research, University of Sydney, Westmead Hospital, Sydney, NSW 2145, Australia. [17]School of Surgery and Pathology, QE11 Medical Centre, M block 2nd Floor, Nedlands, WA 6907, Australia. [18]Southern Health Familial Cancer Centre, Monash Medical Centre, Special Medicine Building, 246 Clayton Rd, Clayton, VIC 3168, Australia. [19]Walter and Eliza Hall Institute, C/o Royal Melbourne Hospital, Grattan Street, Parkville 3050, Australia. [20]Genetic Health Services Victoria, Royal Children's Hospital, Melbourne, VIC 3050, Australia. [21]Kolling Institute of Medical Research, Royal North Shore Hospital, St Leonards, NSW 2065, Australia. [22]Clinical Chemistry, Princess Margret Hospital for Children, Box D184, Perth, WA 6001, Australia. [23]Anatomical Pathology, Prince of Wales Hospital, Randwick 2031 NSW, Australia. [24]Department of Medical Genetics, Women's and Children's Hospital, North Adelaide, SA 5006, Australia. [25]The Peter MacCallum Cancer Centre, Victorian Comprehensive Cancer Centre, Grattan Street, Melbourne 3000, Australia. [26]SA Tissue Pathology, IMVS, Adelaide, SA 5000, Australia. [27]Breast Cancer Laboratory, Walter and Eliza Hall Institute, PO Royal Melbourne, Hospital, Parkville, VIC 3050, Australia. [28]Queensland Institute of Medical Research, Royal Brisbane Hospital, Herston, QLD 4029, Australia. [29]Westmead Institute for Cancer Research, Westmead Millennium Institute, Westmead, NSW 2145, Australia. [30]Department of Surgery, Royal Adelaide Hospital, Adelaide, SA 5000, Australia. [31]Brain and Mind Centre, Camperdown, NSW 2050, Australia. [32]Department of Medicine, Royal Melbourne Hospital, Parkville 3050 Australia. [33]Genetic Services of WA, King Edward Memorial Hospital, 374 Bagot Road, Subiaco, WA 6008, Australia. [34]Epigenetics Unit, Department of Surgery and Oncology, Imperial College London, London W12 0NN, England. [35]Breast Endocrine and Surgical Unit, Royal Adelaide Hospital, North Terrace, SA 5000, Australia. [36]Centre for Genetic Origins of Health and Disease, University of Western Australia, 35 Stirling Highway, Crawley, WA 6009, Australia. [37]Royal Hobart Hospital, GPO Box 1061 L, Hobart, TAS 7001, Australia. [38]Breast Pathology, University of Queensland Centre for Clinical Research, Royal Brisbane and Women's Hospital, Herston, Qld 4029, Australia. [39]Director Surgical Oncology, University of Newcastle, Newcastle Mater Hospital, Waratah, NSW 2298, Australia. [40]Familial Cancer Service, Department of Medicine, Westmead Hospital, Westmead, NSW 2145, Australia. [41]Hereditary Cancer Clinic, Prince of Wales Hospital, Randwick, NSW 2031, Australia. [42]Family Cancer Clinic, St Vincent's Hospital Sydney, Darlinghurst 2010, Australia. [43]Department of Medical Oncology, Peter MacCallum Cancer Centre, Victorian Comprehensive Cancer Centre, Grattan Street, Melbourne, VIC 3000, Australia. [44]Medical Oncology and Clinical Haematology Unit, Western Hospital, Footscray, VIC, Australia. [45]School of Medicine, the University of Notre Dame, Kogarah, NSW 2217, Australia. [46]Department of Pathology, University of Otago, Christchurch New Zealand. [47]Department of Pathology, University of Queensland Medical School, Herston, QLD 4006, Australia. [48]Hunter Family Cancer Service, PO Box 84, Waratah, NSW 2298, Australia. [49]Family Cancer Clinic, Monash Medical Centre, Clayton 3168, Australia. [50]Centre for Epidemiology and Biostatistics, School of Population and Global Health, The University of Melbourne, Carlton, VIC 3053, Australia. [51]The Garvan Institute of Medical Research, The Kinghorn Cancer Centre, Darlinghurst, NSW 2010, Australia. [52]Genetic Health Services Victoria, Royal Children's Hospital, Melbourne, VIC 3050, Australia. [53]The Family Cancer Clinic, Austin Health, Heidelberg, VIC 3084, Australia. [54]Medical Psychology, University of Sydney, Sydney 2006, Australia. [55]University of Queensland, St. Lucia, QLD 4072, Australia. [56]Clinical Geneticist, Royal North Shore Hospital, St Leonards, NSW 2065, Australia. [57]Department of Medical Oncology, Prince of Wales Hospital, Randwick, NSW 2031, Australia. [58]Queensland Clinical Genetic Service, Royal Children's Hospital, Bramston Terrace Herston, Herston, QLD 4020, Australia. [59]Centre for Genetic Education, Prince of Wales Hospital, Randwick, NSW 2031, Australia. [60]Gynaecological Cancer Research, St John of God Subiaco Hospital, 12 Salvado Road, Subiaco, WA 6008, Australia. [61]The University of Queensland Diamantina Institute, Brisbane, QLD 4102, Australia. [62]The University of Queensland, RBWH Campus, Herston, QLD 4029, Australia. [63]Regional Cancer and Blood Services, Auckland City Hospital, Level 1 Building 7, Grafton, Auckland 1023, New Zealand. [64]Medical Psychology Unit, Royal Prince Alfred Hospital, Camperdown, NSW 2204, Australia. [65]Department of Medical Oncology, Westmead Hospital, Westmead, NSW 2145, Australia. [66]Hunter Area Pathology Service, John Hunter Hospital, New Lambton Heights, NSW 2310, Australia. [67]Department of Medical Oncology, The Royal Melbourne Hospital, Parkville, VIC 3050, Australia. [68]Illawarra Cancer Centre, Wollongong Hospital, South Coast Mail Centre, Private Mail Bag 8808, Wollongong, NSW 2521, Australia. [69]Pathology Department, Peter MacCallum Cancer Centre, Victorian Comprehensive Cancer Centre, Grattan Street, Melbourne 3000, Australia. [70]Department of Oncology, St Vincent's Hospital, 41 Victoria Parade, Fitzroy, VIC 3065, Australia. [71]UQ Centre for Clinical Research, University of Queensland, The Royal Brisbane & Women's Hospital Herston, Level 6 Building 71/918, St, Herston 4029, Australia. [72]Breast/Ovarian Cancer Risk Management Clinic, Royal Melbourne Hospital, Parkville, VIC 3050, Australia. [73]The Family Cancer Clinic, Cabrini Hospital, Malvern, VIC 3144, Australia

