## [Peer Review File · Nature Communications]

Reviewers' comments:

Reviewer #1 (Remarks to the Author): Expert in epigenomics

1. Title: "Heritable epimutations" is an oxymoron. The term epimutation refers to changes in DNA methylation (and/or chromatin) that occur independently of genetics, and cause a disease or a specific phenotype (well documented examples are bona fide epimutations that cause Beckwith-Weidemann syndrome, Prader-Willi syndrome, Silver Russel syndrome). So, this can be a very useful and informative term, when applied correctly. However, based on the data in this manuscript, the phenomenon that the authors are actually describing is NOT epimutations. Rather, it is methylation quantitative trait loci (which can be abbreviated as mQTL or meQTL), a well described and well-studied phenomenon that is pervasive in human genomes. It would be a bad mistake and disservice to the field to dilute the useful term epimutations by applying it to the situation that the authors describe in their manuscript. They should use the correct term mQTL.
2. Abstract and main text: Similarly, in their Abstract, the authors state "Mendelian-like inheritance of germline DNA methylation in particular cancer susceptibility genes. We aimed to identify heritable methylation marks associated with breast cancer susceptibility." This sentence precisely describes mQTLs. They should use this well accepted and standard term throughout. In other words, where the phrase "heritable methylation marks" appears, it should first be defined as equivalent to methylation quantitative trait loci, and then be abbreviated as "mQTLs" or "meQTLs" throughout the remaining text.
3. Abstract/study design: the study design is commendable in that it included both a reasonably large "test set" of samples (PBL from breast cancer families), and a "replication set" of samples from a population study.
4. Page 6: "Of the 1,000 most Mendelian methylation marks, 11 of them were associated with breast cancer at the Bonferroni-adjusted p-value threshold of 5×10^{-5} ". This statement again is essentially the definition of mQTLs. That is, loci for which the levels of CpG methylation are genetically determined, by the haplotypes in which the CpGs are embedded. So, "most Mendelian methylation marks" should be stated as "most Mendelian methylation marks, i.e. mQTLs".
5. Page 7: The authors refer to a "Figure 2B", but this reviewer cannot find any figures in the main manuscript file (there are tables in it, but not figures). There are some Supplemental Figures, but it is impossible to know if any of these might correspond to "Figure 2B". This problem of potentially "missing figures" is obviously a major one.
6. Page 7 and overall study: Most crucially, since the authors are studying mQTL's, i.e. "heritable methylation marks", the methylation values at these loci should show correlations with the genotypes of nearby SNPs (most mQTLs are cis-regulated, and all of them that show simple Mendelian inheritance, as in their study, should have this property. Also, it is well established from prior studies that the vast majority of mQTLs are cis-regulated by nearby SNPs located within 1kb of the index CpG). So, if the manuscript is revised and resubmitted, for this reviewer it is essential that the authors provide simple scatter plots of the methylation values for each of their top breast cancer-associated loci (the ones listed in their Table 1), with the methylation values grouped by the genotype of nearby SNPs. In other words, the methylation values for a given CpG (from the 450K Beadchip data, Table 1) plotted separately for all individuals with AA, AB, and BB genotypes. To make these "gold standard" plots, the authors will need to determine SNP genotypes around each of their top-ranked loci, but that is easy and can be done using Illumina 2.5M or 5.0M SNP array data for the same samples, or more cheaply by simple Sanger sequencing of 1kb amplicons centered on each of their 11 top-ranked CpGs. If cost of even the Sanger sequencing is an insurmountable issue, I would be satisfied with seeing such plots from as few as 5 of their top-ranked loci. The data will be very informative, and the results may potentially change the authors' conclusions. It simply has to be done.

Reviewer #2 (Remarks to the Author): Expert in breast cancer genetics

Joo et al have studied Mendelian inheritance of methylation marks and association with breast cancer in 25 extended Australian breast cancer families, using genome wide methylation analysis of 210 blood samples (87 breast cancer cases and 123 unaffected controls). Out of the 1000 most heritable marks, 11 methylation marks were found to associate significantly with breast cancer in the families. In addition, these 11 marks were studied for association with breast cancer in a case-control material of 435 invasive breast cancer cases and their matched controls. Three of these marks were found to associate nominally significantly with breast cancer also in the case-control material. While constitutional methylation has been hypothesized as a mechanism for many inherited diseases it has been little studied in breast cancer so far. The authors have also developed a method for identifying heritable methylation sites that could be of interest in general also for other diseases. This is an interesting and extensive study that brings new information on epimutations as a possible mechanism also for breast cancer risk.

The methylation marks are indicated to have substantial differences between individuals and fall into hyper-, hypo- or hemimethylated groups. It is not very clear in the manuscript whether the 11 marks identified show consistent or different methylation status and association with breast cancer between the families and also compared to the case-control data set. The M-values would be also useful to show. A table compiling these information could be useful.

In the family-based analysis, only p-values are given and not odds ratios for association with breast cancer risk. The authors indicate the ORs in the family-based analyses would be biased which is undoubtedly the case with ascertainment of multiple case families and no adjustment for this ascertainment criterion. However, if the marks were first selected based on Mendelian inheritance in the families where breast cancer is segregating as well and cases have likely been oversampled, would that not lead to inflated test statistics for p-values as well? Would the associations survive adjustment for this? Unbiased odds ratios for the risk would also be more informative for the evaluation of the (biological) significance of the findings.

In table 2, the ORs in the MCCS case-control analysis are similar to low penetrance risk variants in general (0.83-1.26 for the nominally significant marks), suggesting a risk modifying effect rather than a causative role for the disease as such. In the family-based analysis, the risk of breast cancer is indicated to increase with carrier probabilities for all 11 sites. How is this consistent with the risks in the case-control-set (OR 0.83-1.26)? What could be the approximate effect sizes (OR) in the families (see comment above) - falling into low penetrance/modifier category or having a more substantial disease risk? Please discuss the possible significance of the findings in the framework of other breast cancer risk factors or alleles. How would the risk effects detected here compare with those discussed in the introduction?

Altogether, the relationship between hyper/hypo/hemimethylation, direction of the risk effect (risk/protective) and further, effect on the putative target genes is unclear. In the discussion, please elaborate in more detail on this and the methylation status and effects on the expression of the respective genes, rather than dysregulation in general.

The breast cancer risk analyses were adjusted for several risk factors. Are the risk associations similar or different by estrogen receptor status, i.e. in ER positive or ER negative breast cancer for the 11 marks, and specifically, associating with the GREB1 gene "growth regulation by estrogen in breast cancer 1"?

Reviewer #3 (Remarks to the Author): Expert in breast cancer genetics

Reviewer: Paul Pharoah

Summary

This is a clear, well written manuscript reporting a study investigating the association between individual DNA methylation marks (epimutations) in lymphocyte DNA and breast cancer risk. A family based study design is used to identify 11 epimutations with strong statistical evidence of association with breast cancer risk. Overall the findings are novel and reasonably convincing that at least some heritable epimutations are associated with breast cancer risk.

Specific comments

1. As part of the rationale for this study the authors state that heritable epimutations might account for some of the excess familial risk not explained by the known germline genetic variation. This rationale is repeated in the first sentence of the discussion. However, this argument needs further explanation. If an epimutation is truly heritable - passes from one generation to the next - it will presumably be linked to/correlated with nearby germline DNA variation. This DNA variation would also be associated with disease risk and so would account for some of the excess familial risk of disease. The argument is circular. The authors then go on to state that the known example of transgenerational epimutations in mismatch repair genes are in fact linked to nearby cis-acting variants. One would need functional genomic studies to establish whether or not an epimutation was caused by nearby cis-acting variants and whether or not the epimutation itself had any relevant direct functional consequence that resulted in disease risk.

2. The methodology and general approach will not be familiar to most non-specialist readers (such as myself).

It would therefore be helpful to include a main figure summarising the data and analysis for one of the associated epimutations that was also "significant" in the validation nested case-control study in order to aid the reader in understanding the underlying methodology. For example, if I have understood correctly, Figure S4 shows a trimodal distribution of methylation values for cg18584561 with means at approx -4, 0 and 2. I would interpret this as corresponding to three genotypes of a common bi-allelic variant. It is then unclear how this relates to Figure S3. These data (distributions) could perhaps be shown separately for cases and controls from the familial samples (given the association I presume the distributions are different) together with the equivalent panel for Figure S3. Finally the M-value distribution in cases and controls for the validation nested case-control study could be shown.

3. It would be useful to provide the estimated carrier frequency for each of the associated epimutations in the familial samples (by case-control status).

4. If some, or all of these epimutations were confirmed to be associated with breast cancer risk, they would be just like any germline variant only a marker for risk and some causal mechanisms would need to be established to make any claims beyond association.

5. I do not agree that reducing the multiple testing burden increases statistical power in a useful way. The probability that an association that is declared significant at a predetermined threshold is a true positive depends on the statistical power and the prior probability of association. Reducing the number of tests does not alter the prior at all or the power. Whether or not a heritable epimutation is more likely to be associated with risk than a non-heritable epimutation is not known. Some evidence for this could be provided by investigating the association of the least heritable mutations with risk.

6. p10, l198 et seq. The authors state that it is remarkable that three of the eleven associated markers were associated with risk in an independent nested case control study (not cohort study as stated). It is not clear to me why this finding is unexpected. The replication suggests that these epimutations have a population frequency that is sufficient to be detectable in a modest sized case-control study. The authors also speculate that the carrier frequency for some of the epimutations that did not replicate may have been too low. The histograms of the M-values in

cases and controls from the nested case control study ought to be provided in Fig S4 to demonstrate the difference in likely carrier frequencies.

7. Some estimation of the power of the replication study to detect the types of variant identified by the family study should be provided.

8. If the frequency is sufficient to be detectable in the replication study one would expect the epimutation to be correlated with a common DNA variant (SNP) and, as such ought to have been detected in one of several large scale GWAS for breast cancer. Presumably germline genotyping data are available for some or all of these samples and so correlation between the epimutation and nearby SNPs should be evaluated and evidence that these SNPs are indeed associated with risk could be obtained (the authors have easy access to results from multiple BCAC studies).

Discussion

9. The final sentence of the discussion is simply not justified. If heritable epimutations are in cis with DNA variants how will epimutations provide new opportunities for increasing the precision of current risk models or how will they help in developing new strategies for cancer control? I accept that epimutations might offer therapeutic targets, but it would first need to be established that these epimutations are not simply markers of risk - i.e. they have functional consequences that make them valid targets.

Methods

10. The timing of the collection of blood from the familial samples should be stated. In particular, were the case samples collected before or after diagnosis. If the latter is the case then the possibility of reverse causation should be discussed – it would be a potential reason for non-replication in the nested case-control study.

11. A brief explanation of the meaning of the beta- and M-values should be provided in the methods. In addition, because much of the methodology described in this paper will be unfamiliar to the non-specialist reader it would be helpful if an explanation of some of the terms used were provided when first mentioned in the results to avoid the need from switching back and forth between results and methods. E.g. put a brief definition of beta-values, M-values and delta-I in parentheses when they are first mentioned.

Statistical analysis

12. I am not a statistical geneticist, but the statistical approach seems sound and has been explained and justified clearly. Potential biases have been acknowledged and the fact that these do not invalidate the final p-values as a test of association of the epimutation sites of interest noted.

13. Cox PH regression is used to test for association between estimated carrier probability and breast cancer. It would be helpful to show the estimated carrier frequencies in cases and controls.

14. It is stated in the methods (p16, l326) that a Bonferroni correction was used to adjust for multiple testing. However the p-values in Table 1 are unadjusted p-values. It would be more appropriate to state that a Bonferroni corrected threshold was used to determine statistical significance (and to state that threshold).

15. Conditional logistic regression with M-value as the independent variable is used to test for association in the nested case-control study. Given that the underlying model from the family study was Mendelian it would seem more appropriate to use most likely carrier status – at least for those epimutations with a multi-modal distribution with a clear separation between the methylation value peaks (carriers and non-carriers). Presenting the M-values for cases and

controls (see comment # 7) would illustrate this.

Minor comments (very minor for some)

16. Figure S2 is of poor quality/low resolution

17. Figure S3. The total number of samples from the y-axis of the histograms seems to be much less than the total sample size.

18. p6, l113 (typo)" ... of a known SNPs....." (delete a)

19. p7, l127. The statement that HRs could not be calculated should more accurately state that unbiased HRs could not be calculated.

20. p7, l129. It is not very clear to me how Figure S3 shows the estimated effect of the hypothetical genetic variant on the M-values. It would be helpful to explain why the fitted distributions for some epimutations do not seem to fit well (e.g. cg18584561).

20. These authors should realise that "... a number of ...", as used in supplementary statistical methods (p4) could include the number zero. As such it is an unhelpful phrase.

21. The authors ".....wish to thank.....". I wonder then why they do not do so.

RE: NCOMMS-16-20255

Heritable DNA methylation associated with susceptibility to breast cancer.

Thank you for reviewing our manuscript and providing us with the opportunity to respond to the reviewers' comments which we address point by point below.

REVIEWER #1

1. Reviewer comment: *Title: “Heritable epimutations” is an oxymoron. The term epimutation refers to changes in DNA methylation (and/or chromatin) that occur independently of genetics, and cause a disease or a specific phenotype (well documented examples are bona fide epimutations that cause Beckwith-Weidemann syndrome, Prader-Willi syndrome, Silver Russel syndrome). So, this can be a very useful and informative term, when applied correctly. However, based on the data in this manuscript, the phenomenon that the authors are actually describing is NOT epimutations. Rather, it is methylation quantitative trait loci (which can be abbreviated as mQTL or meQTL), a well described and well-studied phenomenon that is pervasive in human genomes. It would be a bad mistake and disservice to the field to dilute the useful term epimutations by applying it to the situation that the authors describe in their manuscript. They should use the correct term mQTL.*

Author response: It has been challenging to find the terminology appropriate for this work. Indeed, many of the methylation marks that we were calling “heritable epimutations” and that are associated with diseases are now linked to genetic variants, yet some others remain independent of known genetic variation. We agree that the term “heritable epimutations” is often used incorrectly and our use of this term in this manuscript did not only refer to the classical situation/definition described above by the reviewer.

The mechanistic explanation for the heritable methylation marks that we describe in this manuscript remains speculative. Indeed, there may be several mechanisms that give rise to methylation marks that are heritable and associated with breast cancer risk. We present the hypothesis that genetic variation may underlie at least some of these heritable methylation marks in the manuscript but the work to explore this hypothesis lies outside the scope of this report (discussed further below).

Thus, without reporting genetic variants linked to our methylation marks, and genetic variants being unlikely to explain all of the heritable methylation marks that we describe in the manuscript (see discussion related to *VTRNA2-1/mir886*). For this reason, we think that “mQTL” is also unsuitable for describing our finding. We understand the reviewers point and have replaced the term “heritable epimutation” with “heritable methylation mark” – which more precisely describes what we have measured and what we are reporting.

We have adjusted the manuscript throughout to address this important issue. Key changes in the text include a paragraph that addresses terminology and how we are applying it in the introduction and further speculation about at least some proportions of the reported heritable methylation marks being mQTLs in the discussion.

2. Reviewer comment: *Abstract and main text: Similarly, in their Abstract, the authors state “Mendelian-like inheritance of germline DNA methylation in particular cancer susceptibility*

genes. We aimed to identify heritable methylation marks associated with breast cancer susceptibility.” This sentence precisely describes mQTLs. They should use this well accepted and standard term throughout. In other words, where the phrase “heritable methylation marks” appears, it should first be defined as equivalent to methylation quantitative trait loci, and then be abbreviated as “mQTLs” or “meQTLs” throughout the remaining text.

Author response: As above, we have reconsidered the terminology used to describe our work and our findings. Throughout the revised manuscript, we use the term “heritable methylation mark” as this is what we sought to identify. We hypothesise and provide additional text for the reader to convey that a proportion of these heritable methylation marks are likely to be due to mQTLs but do not want to label any of the findings with this term until this has been demonstrated.

3. Reviewer comment: *Abstract/study design: the study design is commendable in that it included both a reasonably large “test set” of samples (PBL from breast cancer families), and a “replication set” of samples from a population study.*

Author response: Thank you.

4. Reviewer comment: *Page 6: “Of the 1,000 most Mendelian methylation marks, 11 of them were associated with breast cancer at the Bonferroni-adjusted p-value threshold of 5×10^{-5} ”. This statement again is essentially the definition of mQTLs. That is, loci for which the levels of CpG methylation are genetically determined, by the haplotypes in which the CpGs are embedded. So, “most Mendelian methylation marks” should be stated as “most Mendelian methylation marks, i.e. mQTLs”.*

Author response: Please see the responses to 1 and 2 above.

5. Reviewer comment: *Page 7: The authors refer to a “Figure 2B”, but this reviewer cannot find any figures in the main manuscript file (there are tables in it, but not figures). There are some Supplemental Figures, but it is impossible to know if any of these might correspond to “Figure 2B”. This problem of potentially “missing figures” is obviously a major one.*

Author response: All figures were submitted to the journal/editor. It should be possible to make these available to the reviewer.

6. Reviewer comment: *Page 7 and overall study: Most crucially, since the authors are studying mQTL's, i.e. “heritable methylation marks”, the methylation values at these loci should show correlations with the genotypes of nearby SNPs (most mQTLs are cis-regulated, and all of them that show simple Mendelian inheritance, as in their study, should have this property. Also, it is well established from prior studies that the vast majority of mQTLs are cis-regulated by nearby SNPs located within 1kb of the index CpG). So, if the manuscript is revised and resubmitted, for this reviewer it is essential that the authors provide simple scatter plots of the methylation values for each of their top breast cancer-associated loci (the ones listed in their Table 1), with the methylation values grouped by the genotype of nearby SNPs. In other words, the methylation values for a given CpG (from the 450K Beadchip data, Table 1) plotted separately for all individuals with AA, AB, and BB genotypes. To make these “gold standard” plots, the authors will need to determine SNP genotypes around each of their top-ranked loci, but that is easy and can be done using Illumina 2.5M or 5.0M SNP array data for the same samples, or more cheaply by simple Sanger sequencing of 1kb amplicons centered on each of their 11 top-ranked CpGs. If cost of even the Sanger*

sequencing is an insurmountable issue, I would be satisfied with seeing such plots from as few as 5 of their top-ranked loci. The data will be very informative, and the results may potentially change the authors' conclusions. It simply has to be done.

Author response: The reviewer identifies an extremely important line of investigation. However, the authors do not wish to conduct a quick and limited analysis just to provide some information for this report. This line of investigation requires a comprehensive analysis and it is unlikely to be as straightforward as the reviewer suggests, due to the differences in the frequency of the identified marks, the possible differences in the magnitude of the associated breast cancer risk, phenocopies and the likelihood that at least some of these marks are epimutations (in the strictly defined sense). We are planning a comprehensive analysis to address this question that will be part of a future report (as recognised by reviewer 3 below). We also hope that reporting the findings of our empirically identified *heritable* methylation marks associated with breast cancer risk may stimulate further investigation of this important aspect of the work in the broader research community.

The chromosomal regions on which these marks have been identified have not been associated with breast cancer risk via genome-wide associated studies (information now included in our manuscript), which also suggests that a comprehensive (rather than quick and limited) study of this question is required.

REVIEWER #2

1. Reviewer comment: *Joo et al have studied Mendelian inheritance of methylation marks and association with breast cancer in 25 extended Australian breast cancer families, using genome wide methylation analysis of 210 blood samples (87 breast cancer cases and 123 unaffected controls). Out of the 1000 most heritable marks, 11 methylation marks were found to associate significantly with breast cancer in the families. In addition, these 11 marks were studied for association with breast cancer in a case-control material of 435 invasive breast cancer cases and their matched controls. Three of these marks were found to associate nominally significantly with breast cancer also in the case-control material. While constitutional methylation has been hypothesized as a mechanism for many inherited diseases it has been little studied in breast cancer so far. The authors have also developed a method for identifying heritable methylation sites that could be of interest in general also for other diseases. This is an interesting and extensive study that brings new information on epimutations as a possible mechanism also for breast cancer risk.*

The methylation marks are indicated to have substantial differences between individuals and fall into hyper-, hypo- or hemimethylated groups. It is not very clear in the manuscript whether the 11 marks identified show consistent or different methylation status and association with breast cancer between the families and also compared to the case-control data set. The M-values would be also useful to show. A table compiling these information could be useful.

Author response: The logistic-transformed M-values, which should roughly indicate % methylation levels, for the 11 marks are shown in Supplementary Figure 3. We have now put the histograms of Supplementary Figure 4 on the same scale as Supplementary Figure 3, so that the distributions of methylation for the family analysis can be directly compared to that of the case-control analysis. We have also added a table (Supplementary Table 2) showing the number of hypo-, hemi- and hypermethylated cases and controls for the 11 marks.

2. Reviewer comment: *In the family-based analysis, only p-values are given and not odds ratios for association with breast cancer risk. The authors indicate the ORs in the family-*

based analyses would be biased which is undoubtedly the case with ascertainment of multiple case families and no adjustment for this ascertainment criterion. However, if the marks were first selected based on Mendelian inheritance in the families where breast cancer is segregating as well and cases have likely been oversampled, would that not lead to inflated test statistics for p-values as well?

Author response: The p-value is the probability of observing data as or more extreme as the observed data, under the assumption that the null hypothesis is true. Under the null hypothesis, there is no association between breast cancer and the carrier probabilities for the probe, so oversampling for cases does not affect the distribution of the test statistic, hence the p-value is unbiased. This is analogous to the way that oversampling cases in a case-control study does not bias the p-value because even though cases are oversampled, there are no constraints on the exposure. Marks were first selected based on Mendelian inheritance, independently of case status.

3. Reviewer comment: *Would the associations survive adjustment for this? Unbiased odds ratios for the risk would also be more informative for the evaluation of the (biological) significance of the findings.*

Author response: Unfortunately, there is no conventional way to adjust for the clinic-based ascertainment of the families. Even methods that might be applied for mQTLs would require the relevant genetic variant to be measured. We therefore used an independent and population-based dataset, from the MCCS, to estimate unbiased ORs. Note that we expect aberrant methylation at most of these probes to be rare (the aim of the study is to find heritable factors with high enough risks to explain multiple-case families, and such factors must be rare) so we expect our power to detect these marks to be low in a population-based study (just as power is low in a GWAS to detect a very rare yet “high-risk” *BRCA1* mutation). This is likely to explain why only some of the heritable methylation marks were associated with breast cancer risk in the MCCS.

4. Reviewer comment: *In table 2, the ORs in the MCCS case-control analysis are similar to low penetrance risk variants in general (0.83-1.26 for the nominally significant marks), suggesting a risk modifying effect rather than a causative role for the disease as such. In the family-based analysis, the risk of breast cancer is indicated to increase with carrier probabilities for all 11 sites. How is this consistent with the risks in the case-control-set (OR 0.83-1.26)?*

Author response:

On the direction of the effects, a genetic variant can increase or decrease methylation at a site, and a change in the level of methylation at a site can cause breast cancer, regardless of the direction of this change. So an $OR > 1$ for the M-values of a particular site (in the MCCS) and an $HR > 1$ for the carrier probabilities (in the family-based analyses) could occur if a genetic variant increases M-values at the site and this causes breast cancer. Similarly, an $OR < 1$ for M-values and an $HR > 1$ for the carrier probabilities could occur if a genetic variant decreases M-values at the site and this causes breast cancer. For example, for cg03916490 (near *C7orf50*), Supp. Fig. 3 shows that carriers generally have lower M-values than non-carriers, so since carriers have higher risks than non-carriers, we would expect $OR < 1$, as observed.

On the size of the effects, note that the reported ORs for each mark in the MCCS analysis are the estimated ORs per standard deviation (SD). In a population-based study like the MCCS, we would only expect a small amount of variation in the M-values of these probes. However,

a rare genetic variant that causes aberrant methylation at the probe could have a large effect on its methylation levels, and this would correspond to a large relative risk for carriers. For example, cg01741999 (near PNKD) has an OR per SD of 1.26, so if a genetic variant changes methylation by 5 SDs then the variant would have an OR of 3.2 ($=1.26^5$). In other words, more substantial variation due to a rare, heritable shift in methylation values may be associated with much larger increases in risk in multiple-case families.

5. Reviewer comment: *What could be the approximate effect sizes (OR) in the families (see comment above) - falling into low penetrance/modifier category or having a more substantial disease risk? Please discuss the possible significance of the findings in the framework of other breast cancer risk factors or alleles. How would the risk effects detected here compare with those discussed in the introduction?*

Author response:

Further work is required to estimate the effect sizes. We can only hypothesise. See our response to Comment 4 above, where we discuss extrapolating effect sizes from the population-based analyses to the family-based analyses.

6. Reviewer comment: *Altogether, the relationship between hyper/hypo/hemimethylation, direction of the risk effect (risk/protective) and further, effect on the putative target genes is unclear. In the discussion, please elaborate in more detail on this and the methylation status and effects on the expression of the respective genes, rather than dysregulation in general.*

Author response: With respect, we think the associations between the M-values and breast cancer are clearly laid out in Table 2 (as we mention when responding to the comment 3 above, we can only estimate effect sizes using the population-based MCCS data). To clarify this, we have now also added Supplementary Table 2, which gives a contingency table and p-value for the association between breast cancer and methylation β -values categorised into three groups (hypo-, hemi- and hypermethylated) for each probe. We also think we have been clearer about the direction of effect of the hypothetical genetic variants on breast cancer risk (and as noted above, we can't estimate the size of the effect in the family-based analyses). For example, we stated in the manuscript that "the risk of breast cancer increased with carrier probabilities for all 11 sites" (line 144). In addition, the effect of the hypothetical genetic variant on the M-values of each site is precisely given both graphically and in tabular form, as we say "the estimated effect of the hypothetical genetic variant on the M-values of each site can be seen from Supplementary Figure S3 or Supplementary Table 1" (line 146). We have attempted to make this clearer by explaining it in more detail (line 148-152).

7. Reviewer comment: The breast cancer risk analyses were adjusted for several risk factors. Are the risk associations similar or different by estrogen receptor status, i.e. in ER positive or ER negative breast cancer for the 11 marks, and specifically, associating with the GREB1 gene "growth regulation by estrogen in breast cancer 1"?

Author response: This is an interesting suggestion and we tested for an association with ER status. Only one methylation mark was associated with ER status ($p < 0.05$), which may be due to chance (and was not cg18584561 at GREB1). We have included this in the manuscript (line 192-195).

REVIEWER #3

Summary

This is a clear, well-written manuscript reporting a study investigating the association between individual DNA methylation marks (epimutations) in lymphocyte DNA and breast cancer risk. A family based study design is used to identify 11 epimutations with strong statistical evidence of association with breast cancer risk. Overall the findings are novel and reasonably convincing that at least some heritable epimutations are associated with breast cancer risk.

Specific comments

1. Reviewer comment: *1. As part of the rationale for this study the authors state that heritable epimutations might account for some of the excess familial risk not explained by the known germline genetic variation. This rationale is repeated in the first sentence of the discussion. However, this argument needs further explanation. If an epimutation is truly heritable - passes from one generation to the next - it will presumably be linked to/correlated with nearby germline DNA variation. This DNA variation would also be associated with disease risk and so would account for some of the excess familial risk of disease. The argument is circular. The authors then go on to state that they know example of transgeneration epimutations in mismatch repair genes are in fact linked to nearby cis-acting variants. One would need functional genomic studies to establish whether or not an epimutation was caused by nearby cis-acting variants and whether or not the epimutation itself had any relevant direct functional consequence that resulted in disease risk.*

Author response: We are studying multiple-case families with no known cause of breast cancer, and the rationale for our study is simply that it would be beneficial to find a cause. Even if a heritable methylation mark is caused by a mutation in an unknown gene, or is just associated with an unknown gene in the way the reviewer suggested, then we think identifying that mark is clearly worthwhile, because it will probably help to identify the unknown gene and it might help to identify a mechanism. Further, *cis* or *trans* acting genetic variants responsible for our methylation changes are likely to be situated anywhere in the genome and some of current genomic techniques (e.g. exome-seq, SNP arrays) are likely to miss a large fraction of the genome, especially intergenic regions. Hence, it may be more effective to measure methylation levels.

On the last point, we agree (our text is consistent with this) and as discussed below, a comprehensive analysis of these marks, including functional genomic studies, is being planned and further work may be stimulated via the publication of this report.

2. Reviewer comment: *The methodology and general approach will not be familiar to most non-specialist readers (such as myself). It would therefore be helpful to include a main figure summarising the data and analysis for one of the associated epimutations that was also “significant” in the validation nested case-control study in order to aid the reader in understanding the underlying methodology. For example, if I have understood correctly, Figure S4 shows a trimodal distribution of methylation values for cg18584561 with means at approx -4, 0 and 2. I would interpret this as corresponding to three genotypes of a common bi-allelic variant. It is then unclear how this relates to Figure S3. These data (distributions) could perhaps be shown separately for cases and controls from the familial samples (given the association I presume the distributions are different) together with the equivalent panel for Figure S3. Finally the M-value distribution in cases and controls for the validation nested case-control study could be shown.*

Author response: Supplementary Figures 3 and 4 are now presented on the same scale for clearer comparison. In Supplementary Figures S4, we now present the distributions categorically by separating cases and controls, as suggested. We also have added a new figure illustrating our analytical approach (**Figure 3**).

3. Reviewer comment: *It would be useful to provide the estimated carrier frequency for each of the associated epimutations in the familial samples (by case-control status).*

Author response: We now provide this data in Supplementary Table 3. The low carrier probabilities for some probes are presumably due to the very low prior (equal to 0.02, which is twice the population allele frequency) that was assumed for the probability for carrying the variant, and because at most one branch of the family can carry the variant (we assumed this in our analysis, but it also follows approximately from the rareness of the variant) so most people within each family will be “non-carriers”.

4. Reviewer comment: *If some, or all of these epimutations were confirmed to be associated with breast cancer risk, they would be just like any germline variant only a marker for risk and some causal mechanisms would need to be established to make any claims beyond association.*

Author response: Yes, we agree, we are reporting heritable methylation marks associated with breast cancer risk in this manuscript. Our text is consistent with this intention and consistent with our responses to reviewers 1 and 2.

5. Reviewer comment: *I do not agree that reducing the multiple testing burden increases statistical power in a useful way. The probability that an association that is declared significant at a predetermined threshold is a true positive depends on the statistical power and the prior probability of association. Reducing the number of tests does not alter the prior at all or the power. Whether or not a heritable epimutation is more likely to be associated with risk than a non-heritable epimutation is not known. Some evidence for this could be provided by investigating the association of the least heritable mutations with risk.*

Author response: Thank you for comment. We agree with that screening out non-heritable probes will not necessarily increase our power to detect probes that are associated with the risk of breast cancer. However, (almost tautologically) it will increase the prior probability that a mark taken forward for association testing is associated with *heritable* breast cancer (i.e. breast cancer caused by a heritable factor). Therefore, the screening step will increase our power to detect probes that are associated with the risk of heritable breast cancer. This is the main aim of our study, so it is the context for our claim about improving power by selecting the most heritable methylation marks. However, we should have been more explicit about this, and we have now amended the manuscript to make it clear that our claims about power only apply to the identification of heritable methylation marks associated with breast cancer risk.

6. Reviewer comment: *p10, l198 et seq. The authors state that it is remarkable that three of the eleven associated markers were associated with risk in an independent nested case control study (not cohort study as stated). It is not clear to me why this finding is unexpected. The replication suggests that these epimutations have a population frequency that is sufficient to be detectable in a modest sized case-control study. The authors also speculate that the carrier frequency for some of the epimutations that did not replicate may have been too low.*

The histograms of the M-values in cases and controls from the nested case control study ought to be provided in Fig S4 to demonstrate the difference in likely carrier frequencies.

Author response: Thank you, we have edited the text to appropriately name the study design as a nested case control study (with the MCCS) and the distributions for the MCCS cases and controls are presented separately in Supplementary Figures 4. We have also revised the discussion to address other points above (line 271-284).

7. Reviewer comment: *Some estimation of the power of the replication study to detect the types of variant identified by the family study should be provided.*

Author response: Before doing the MCCS analysis, we did not have any estimates of the true standard deviations (SDs) of the β -values for each probe, or of the true differences between the β -values of cases and controls. Therefore, we can only provide post-hoc power calculations based on the observed SDs and beta-value differences, which are likely to be biased. Most probes had observed beta-value SDs of approximately 0.2 and observed beta-value differences between cases and controls of less than 0.01, so post-hoc power calculations show that these probes had less than 11% chance to replicate. The 3 probes that replicated each had roughly 60% (post-hoc) chance of replicating, based on their observed SDs and beta-value differences, though again that post-hoc power calculation are usually biased. For these reasons, we have decided not to include this in the manuscript.

8. Reviewer comment: *If the frequency is sufficient to be detectable in the replication study one would expect the epimutation to be correlated with a common DNA variant (SNP) and, as such ought to have been detected in one of several large scale GWAS for breast cancer. Presumably germline genotyping data are available for some or all of these samples and so correlation between the epimutation and nearby SNPs should be evaluated and evidence that these SNPs are indeed associated with risk could be obtained (the authors have easy access to results from multiple BCAC studies).*

Author response: As discussed in responses to Reviewer #1, we do not have any evidence that these heritable methylation marks are at or close to loci that have been identified to be associated with breast cancer risk via genome-wide association studies. There is some published information available for the three probes at *VTRNA2-1* that is now included in our report. See line 222.

9. Reviewer comment: Discussion *The final sentence of the discussion is simply not justified. If heritable epimutations are in cis- with DNA variants how will epimutations provide new opportunities for increasing the precision of current risk models or how will they help in developing new strategies for cancer control? I accept that epimutations might offer therapeutic targets, but it would first need to be established that these epimutations are not simply markers of risk - i.e. they have functional consequences that make them valid targets.*

Author response: As discussed above, we present the heritable methylation marks associated with breast cancer risk. This is the outcome of applying the method that we describe in the manuscript. While at least some of these methylation marks are likely due to underlying variation in genetic sequence, we do not have any data to demonstrate this. We have no evidence to suggest that we have re-identified a genetic risk factor (i.e. genetic variation that is already known to be associated with breast cancer risk) but even if we had then we think

the last sentence is justified. That is, our data are likely to provide new information for risk prediction models, even if it is new (i.e. currently unknown) genetic information.

10. Reviewer comment: Methods *The timing of the collection of blood from the familial samples should be stated. In particular, were the case samples collected before or after diagnosis. If the latter is the case then the possibility of reverse causation should be discussed – it would be a potential reason for non-replication in the nested case-control study.*

Author response: Two thirds of the bloods collected for affected members of the multiple-case families were collected after breast cancer diagnosis. The possibility of reverse causation is thus relevant to our report and has been included in the discussion.

11. Reviewer comment: *11. A brief explanation of the meaning of the beta- and M-values should be provided in the methods. In addition, because much of the methodology described in this paper will be unfamiliar to the non-specialist reader it would be helpful if an explanation of some of the terms used were provided when first mentioned in the results to avoid the need from switching back and forth between results and methods. E.g. put a brief definition of beta-values, M-values and delta-l in parentheses when they are first mentioned.*

Author response: We have included the suggested text in the manuscript.

12. Reviewer comment: Statistical analysis *I am not a statistical geneticist, but the statistical approach seems sound and has been explained and justified clearly. Potential biases have been acknowledged and the fact that these do not invalidate the final p-values as a test of association of the epimutation sites of interest noted.*

Author response: Thank you.

13. Reviewer comment: *Cox PH regression is used to test for association between estimated carrier probability and breast cancer. It would be helpful to show the estimated carrier frequencies in cases and controls.*

Author response: This data is now included in Supplementary Table 3 (see response to comment 3).

14. Reviewer comment: *It is stated in the methods (p16, l326) that a Bonferroni correction was used to adjust for multiple testing. However the p-values in Table 1 are unadjusted p-values. It would be more appropriate to state that a Bonferroni corrected threshold was used to determine statistical significance (and to state that threshold).*

Author response: We have changed our phrasing in the way suggested, thanks.

15. Reviewer comment: *Conditional logistic regression with M-value as the independent variable is used to test for association in the nested case-control study. Given that the underlying model from the family study was Mendelian it would seem more appropriate to use most likely carrier status – at least for those epimutations with a multi-modal distribution with a clear separation between the methylation value peaks (carriers and non-carriers). Presenting the M-values for cases and controls (see comment # 7) would illustrate this.*

Author response: To calculate the carrier probabilities of the MCCS cases and controls, we must treat each MCCS participant as a singleton family, because we do not have any data on their families. Therefore, their carrier probabilities are just transformed versions of the M-values, where the transforming function depends on parameters estimated in the family analyses (which might not be exactly right for the MCCS). We think it is cleaner to present the MCCS results based on M-values rather than carrier probabilities in the main manuscript. However, we have added the results based on the carrier probabilities to Supplementary Table 5 (and we have separated Supplementary Figure 4 into cases and controls, as mentioned above). Also, we have now added Table 3, which uses MCCS data to estimate the associations between breast cancer and categorised M-values.

Minor comments (very minor for some)

16. Reviewer comment: *Figure S2 is of poor quality/low resolution*

Author response: We have replaced this figure by a higher quality version.

17. Reviewer comment: *Figure S3. The total number of samples from the y-axis of the histograms seems to be much less than the total sample size.*

Author response: The histograms showed probability densities rather than frequencies. However, influenced by the reviewer's comment, we have redrawn these figures presenting the frequencies as we think this is a more appropriate presentation.

18. Reviewer comment: *p6, l113 (typo) " ... of a known SNPs..... " (delete a*

Author response: This error has been corrected, thanks.

19. Reviewer comment: *p7, l127. The statement that HRs could not be calculated should more accurately state that unbiased HRs could not be calculated.*

Author response: We agree and have made the corresponding change.

20. Reviewer comment: *p7, l129. It is not very clear to me how Figure S3 shows the estimated effect of the hypothetical genetic variant on the M-values. It would be helpful to explain why the fitted distributions for some epimutations do not seem to fit well (e.g. cg18584561).*

Author response: We have now added two examples to the manuscript to help the reader interpret Supplementary Figure 3 (and Supplementary Table 1). For probe cg06536614, carriers are hemi-methylated and non-carriers are hypo-methylated, which is consistent with the hypothetical genetic variant being *cis*-acting, similar to the MLH1 example of (Hitchins et al., 2011). For probe cg18584561, carriers are hypo-methylated and non-carriers have generally higher methylation levels but these are spread over a wide range. The remaining probes are similar to one of these two examples, except for a few probes where non-carriers seem to have tightly constrained methylation values while carriers seem to have a wide range of methylation values.

20. Reviewer comment: *These authors should realise that "... a number of ...", as used in supplementary statistical methods (p4) could include the number zero. As such it is an unhelpful phrase.*

Author response: We have edited the text to address this comment.

21. Reviewer comment: *The authors “.....wish to thank.....”. I wonder then why they do not do so.*

Author response: Thank you, we have removed “wish to” from the text.

Reviewers' comments:

Reviewer #1 (Remarks to the Author):

In this revised paper the authors have clarified their use of the term "heritable epimutation, and appropriately added the term "mQTL", to describe loci in which the methylation status tracks with the genotypes of nearby SNPs.

There is some new discussion of findings regarding CpG methylation at the VTRNA2 locus, which the authors postulate might reflect a combination of genomic imprinting (as described in two reports in the prior literature), and either an mQTL-like effect or a true epimutation. This section has some mistakes in assigning specific references to specific findings (#2 below), and it also will need to be scientifically clarified with two obvious and easy to perform tests "at the bench" (#1 and #3 below). I hope that these specific comments will allow the authors to clarify these key points in their interesting and potentially valuable study.

1. Genomic imprinting of VTRNA2 has been shown in two prior studies. Specifically, Paliwal et al (PLOS Genetics, 2013) found preferential methylation of the maternal allele at this locus in about 80 percent of human tissue samples, and Romanelli et al (Epigenetics, 2014) made the same finding of preferential methylation of the maternal allele in about 75 percent of their samples. So, does this locus show imprinting in the authors' sample set? This key point seems not be directly tested in the manuscript. Since the authors' study design is family-based - doing an imprinting analysis is easy. They should run bisulfite sequencing on trios of mother, father and offspring from some of their families, exactly as has been done in the two prior studies, and determine whether the maternal allele is more frequently hypermethylated than the paternal allele. This type of test would not require all 25 families - 10 trios would be sufficient to arrive at a clear answer.

2. The following sentences and references (lines 206-216) are a bit "jumbled up", and should be corrected as indicated:

"Hypomethylation at this promoter, suggestive of loss of imprinting, occurs systematically in specific individuals in diverse populations, at least partially due to periconceptional environment and is stable for at least 10 years (ref 32 - incorrect reference number - it should be Silver et al. 2015, not Paliwal et al. 2013). Silver et al. (2015) noted that VTRNA2-1 exhibits all the hallmarks of "metabolic imprinting" and is likely to be a determinant of cancer risk (32, 30))(both of these reference numbers are incorrect - should be Silver et al 2015, which is written by name in the main text but seems not be included in the actual reference list). Here, we have shown that methylation at the VTRNA2-1 promoter is also associated with heritable breast cancer risk that is measurable in DNA extracted from blood. All 210 DNAs included in this study had hemi- or hypomethylation across all 6 CpG probes at the VTRNA2-1 locus (Figure 2) indicating potential allele-specific DNA methylation (ASM). ASM at this locus has been reported previously by studies utilizing clonal bisulfite sequencing of multiple tissue types 31, 32 (here the reference numbers ARE correct). These studies were unsuccessful in identifying any nearby genetic variation that segregated with this allelic methylation pattern (more accurate would be to say - "However, these studies did not explore nearby genetic variation that might be superimposed on imprinting to influence the allelic methylation pattern").

3. Following up on #2 above, if the authors are correct and the VTRNA2 locus in fact shows an "influence of nearby genetic variation" on its CpG methylation, then the obvious question is -which nearby SNP genotype confers relative hypomethylation and which one confers relative hypermethylation? The answer would emerge either from the bisulfite sequencing that I have suggested in #1 above (examining SNPs in the PCR products that are not destroyed by the

bisulfite conversion), or from standard Sanger sequencing of the same samples.

If, on the other hand, the authors believe that their observations for VTRNA2 do NOT reflect an mQTL-type effect, and instead reflect true epimutations, then why do they have the sentence above about the influence of "nearby genetic variation"?

In summary, the situation regarding the VTRNA2 locus is interesting, but it needs to be clarified. First, is this gene imprinted in the authors' sample set? Second, are the authors' observations for this gene due to an mQTL-type effect superimposed on parental imprinting? Or, do their data reflect true epimutations, superimposed on parental imprinting? The additional work that I have suggested would clarify the situation.

Reviewer #3 (Remarks to the Author):

This is a re-review of a manuscript modified in response to comments of the reviewers.

Most of the comments have been addressed appropriately, however, I remain unconvinced by the response to my original comment #8

"If the frequency is sufficient to be detectable in the replication study one would expect the epimutation to be correlated with a common DNA variant (SNP) and, as such ought to have been detected in one of several large scale GWAS for breast cancer."

The authors respond by stating that the epimutations are not close to known breast susceptibility loci. However, as is well known, there are likely to be many breast cancer susceptibility alleles that do not reach nominal genome-wide significance. The authors ought to have access to the BCAC data that would enable to address this. If there is no evidence for nearby germline genetic association then the authors need to discuss why not. Possible explanations include epimutation result is a false positive or genetic determinants of epimutation are distal to the epimutation (could be empirically tested with data available to the authors).

We were pleased to read that our responses to the reviews of our paper clarified the majority of issues raised. Below please find our response to the remaining queries.

Reviewer #1

Reviewer comment: *There is some new discussion of findings regarding CpG methylation at the VTRNA2 locus, which the authors postulate might reflect a combination of genomic imprinting (as described in two reports in the prior literature), and either an mQTL-like effect or a true epimutation. This section has some mistakes in assigning specific references to specific findings (#2 below), and it also will need to be scientifically clarified with two obvious and easy to perform tests "at the bench" (#1 and #3 below). I hope that these specific comments will allow the authors to clarify these key points in their interesting and potentially valuable study.*

1. Genomic imprinting of VTRNA2 has been shown in two prior studies. Specifically, Paliwal et al (PLOS Genetics, 2013) found preferential methylation of the maternal allele at this locus in about 80 percent of human tissue samples, and Romanelli et al (Epigenetics, 2014) made the same finding of preferential methylation of the maternal allele in about 75 percent of their samples. So, does this locus show imprinting in the authors' sample set? This key point seems not be directly tested in the manuscript. Since the authors' study design is family-based - doing an imprinting analysis is easy. They should run bisulfite sequencing on trios of mother, father and offspring from some of their families, exactly as has been done in the two prior studies, and determine whether the maternal allele is more frequently hypermethylated than the paternal allele. This type of test would not require all 25 families - 10 trios would be sufficient to arrive at a clear answer.

Author response: We have tested for imprinting at the VTRNA2-1 locus by performing clonal bisulfite sequencing on trios (mother, father and child). We performed this on 8 trios and included additional siblings when possible. In total, we sequenced 18 siblings using the assay designed by Paliwal et al (PLOS Genetics, 2013), which encompassed 10 CpG sites downstream of VTRNA2-1 and rs2346019. Consistent with the findings of Paliwal et al and Romanelli et al, we observed strong hypermethylation of the maternally inherited allele, confirming the maternal imprinting of this locus. We found complete loss of methylation in one child whose 3 other siblings retained the methylation in the maternal allele. We have presented this data in a new Supplementary Figure 4.

2. The following sentences and references (lines 206-216) are a bit "jumbled up", and should be corrected as indicated: "Hypomethylation at this promoter, suggestive of loss of imprinting, occurs systematically in specific individuals in diverse populations, at least partially due to periconceptual environment and is stable for at least 10 years (ref 32 - incorrect reference number - it should be Silver et al. 2015, not Paliwal et al. 2013). Silver et al. (2015) noted that VTRNA2-1 exhibits all the hallmarks of "metabolic imprinting" and is likely to be a determinant of cancer risk (32, 30))(both of these reference numbers are incorrect - should be Silver et al 2015, which is written by name in the main text but seems not be included in the actual reference list). Here, we have shown that methylation at the VTRNA2-1 promoter is also associated with heritable breast cancer risk that is measurable in DNA extracted from blood. All 210 DNAs included in this study had hemi- or hypomethylation across all 6 CpG probes at the VTRNA2-1 locus (Figure 2) indicating potential allele-specific DNA methylation (ASM). ASM at this locus has been reported

previously by studies utilizing clonal bisulfite sequencing of multiple tissue types 31, 32 (here the reference numbers ARE correct). These studies were unsuccessful in identifying any nearby genetic variation that segregated with this allelic methylation pattern (more accurate would be to say - "However, these studies did not explore nearby genetic variation that might be superimposed on imprinting to influence the allelic methylation pattern").

Author response: These errors were introduced during the process of manuscript editing and are now corrected. Thank you for alerting us to these.

3. Following up on #2 above, if the authors are correct and the VTRNA2 locus in fact shows an "influence of nearby genetic variation" on its CpG methylation, then the obvious question is -which nearby SNP genotype confers relative hypomethylation and which one confers relative hypermethylation? The answer would emerge either from the bisulfite sequencing that I have suggested in #1 above (examining SNPs in the PCR products that are not destroyed by the bisulfite conversion), or from standard Sanger sequencing of the same samples.

Author response: We do not make any categorical statement about VTRNA2-1 being influenced by nearby genetic variation – it remains a possibility but we do not conclude that this is the case in our text. We have incorporated new data demonstrating maternal imprinting at this locus but we have no evidence of genetic influence from the bisulfite sequencing or the work described below in response to reviewer 3.

If, on the other hand, the authors believe that their observations for VTRNA2 do NOT reflect an mQTL-type effect, and instead reflect true epimutations, then why do they have the sentence above about the influence of "nearby genetic variation"?

Author response: Both possibilities remain plausible. We have no evidence to suggest that genetic variation influences the methylation status at VTRNA2-1 but it remains possible that genetic variation more distant to the region we have searched could be involved.

In summary, the situation regarding the VTRNA2 locus is interesting, but it needs to be clarified. First, is this gene imprinted in the authors' sample set? Second, are the authors' observations for this gene due to an mQTL-type effect superimposed on parental imprinting? Or, do their data reflect true epimutations, superimposed on parental imprinting? The additional work that I have suggested would clarify the situation.

Author response: This additional work has been done and is now described in our manuscript.

Reviewer #3

Most of the comments have been addressed appropriately, however, I remain unconvinced by the response to my original comment #8.

"If the frequency is sufficient to be detectable in the replication study one would expect the epimutation to be correlated with a common DNA variant (SNP) and, as such ought to have been detected in one of several large scale GWAS for breast cancer."

The authors respond by stating that the epimutations are not close to known breast susceptibility loci. However, as is well known, there are likely to be many breast cancer susceptibility alleles that do not reach nominal genome-wide significance. The authors ought to have access to the BCAC data that would enable to address this. If there is no evidence for nearby germline genetic association then the authors need to discuss why not. Possible explanations include epimutation result is a false positive or genetic determinants of epimutation are distal to the epimutation (could be empirically tested with data available to the authors).

Author response: Using available BCAC data, we have searched the genomic regions 1kb upstream and downstream of each methylation mark (or cluster in the case of VTRNA2-1) to identify variants (genotyped or imputed to 1000 genomes) with weaker (than genome-wide significance) evidence of association with breast cancer risk. Three of the 24 regions contain common genetic variants with p-values for association with breast cancer below 0.05 (all ≥ 0.001 , unadjusted for multiple testing). Only one of these regions was associated with breast cancer risk in the MCCS. This region at *GREB1* (cg18584561) was of particular note as it had eight common variants (in linkage disequilibrium) associated with breast cancer risk at $p = 0.05 - 0.02$.

iCOGS (SNP array) data and methylation array data was available for 251 MCCS participants (231 cases and 20 controls). There was a very strong linear association between methylation at cg18584561 and the eight common variants at this region ($P=1 \times 10^{-65} - 1 \times 10^{-71}$). Therefore, breast cancer risk associated with methylation at cg18584561 is likely to be due to this underlying genetic variation. We have included this information (based on the analysis of iCOGS data in the MCCS study) in the manuscript and provided boxplots showing genotypes at the 8 proximal common genetic variants and Beta methylation values at *GREB1* (cg18584561) in Supplementary Fig 6.

These findings are consistent with our discussion – that there are likely to be several underlying biological explanations of heritable methylation marks, including mQTLs. However, for the majority of CpGs identified in our study (21/24), there is no evidence of breast cancer association with nearby common genetic variants, suggesting that at least some are not mQTLs.

Additional revision:

During the revision of our manuscript we noted a single coding error in the calculation of the carrier probabilities. This has now been corrected and the code thoroughly re-checked.

Correction of the code changed the p-values for association with breast cancer. After correction, ten of the eleven originally identified CpGs are still associated with breast cancer at $P < 0.05$ (cg04912316, FAM100B is no longer significant). Fourteen additional CpGs were identified as being associated with breast cancer risk. The manuscript has been updated throughout to report 24 methylation marks associated with breast cancer risk and all tables and figures have been updated.

REVIEWERS' COMMENTS:

Reviewer #1 (Remarks to the Author):

The authors have systematically and successfully addressed my initial requests for clarifications and additional data (much in the Supplementary Material), and the manuscript is now improved.